# A Retro-Aldol Reaction Prompted the Evolvability of a Phosphotransferase System for the Utilization of a Rare Sugar

Yunhye Joo,[a] Jae-Yoon Sung,[a] Sun-Mi Shin,[b*] Sun Jun Park,[c,d] Kyoung Su Kim,[a] Ki Duk Park,[c,d] Seong-Bo Kim,[e] Dong-Woo Lee[a]

[a]Department of Biotechnology, Yonsei University, Seoul, Republic of Korea
[b]School of Applied Biosciences, Kyungpook National University, Daegu, Republic of Korea
[c]Brain Science Institute, Korea Institute of Science & Technology (KIST), Seoul, Republic of Korea
[d]Division of Bio-Medical Science & Technology, KIST School, Korea University of Science and Technology, Seoul, Republic of Korea
[e]Bio-Living Engineering Major, Global Leaders College, Yonsei University, Seoul, Republic of Korea

Yunhye Joo and Jae-Yoon Sung contributed equally to this work. Author order was determined by drawing straws.

**ABSTRACT** The evolution of the bacterial phosphotransferase system (PTS) linked to glycolysis is dependent on the availability of naturally occurring sugars. Although bacteria exhibit sugar specificities based on carbon catabolite repression, the acquisition and evolvability of the cellular sugar preference under conditions that are suboptimal for growth (e.g., environments rich in a rare sugar) are poorly understood. Here, we generated *Escherichia coli* mutants via a retro-aldol reaction to obtain progeny that can utilize the rare sugar D-tagatose. We detected a minimal set of adaptive mutations in the D-fructose-specific PTS to render *E. coli* capable of D-tagatose utilization. These *E. coli* mutant strains lost the tight regulation of both the D-fructose and *N*-acetyl-galactosamine PTS following deletions in the binding site of the catabolite repressor/activator protein (Cra) upstream from the *fruBKA* operon and in the *agaR* gene, encoding the *N*-acetylgalactosamine (GalNAc) repressor, respectively. Acquired D-tagatose catabolic pathways then underwent fine-tuned adaptation via an additional mutation in 1-phosphofructose kinase to adjust metabolic fluxes. We determined the evolutionary trajectory at the molecular level, providing insights into the mechanism by which enteric bacteria evolved a substrate preference for the rare sugar D-tagatose. Furthermore, the engineered *E. coli* mutant strain could serve as an *in vivo* high-throughput screening platform for engineering non-phosphosugar isomerases to produce rare sugars.

**IMPORTANCE** Microorganisms generate energy through glycolysis, which might have preceded a rapid burst of evolution, including the evolution of cellular respiration in the primordial biosphere. However, little is known about the evolvability of cellular sugar preferences. Here, we generated *Escherichia coli* mutants via a retro-aldol reaction to obtain progeny that can utilize the rare sugar D-tagatose. Consequently, we identified mutational hot spots and determined the evolutionary trajectory at the molecular level. This provided insights into the mechanism by which enteric bacteria evolved substrate preferences for various sugars, accounting for the widespread occurrence of these taxa. Furthermore, the adaptive laboratory evolution-induced cellular chassis could serve as an *in vivo* high-throughput screening platform for engineering tailor-made non-phosphorylated sugar isomerases to produce low-calorigenic rare sugars showing antidiabetic, antihyperglycemic, and antitumor activities.

**KEYWORDS** phosphotransferase system, sugar preference, adaptive laboratory evolution, rare sugar, D-tagatose

Address correspondence to Dong-Woo Lee, leehicam@yonsei.ac.kr.

*Present address: Sun-Mi Shin, Institute of Biotechnology, CJ CheilJedang Corp., Suwon, South Korea.

The authors declare no conflict of interest.

In early life, autotrophs might have provided their cell mass as viable nutrients to heterotrophs (1). Sequential ecosystem changes triggered the explosive diversification of

catabolic pathways (2, 3) and adaptation to available nutritional sources (4–6). Among early systems for carbohydrate metabolism, glycolysis efficiently produces ATP for sustaining both autotrophs and heterotrophs (7, 8). Most bacteria possess one or several phosphotransferase systems (PTS) for different sugars, producing phosphorylated sugars fed into oxygen-free glycolysis, which is followed by aerobic and/or anaerobic respiration for ATP production (9–11). Extant heterotrophs utilize naturally occurring carbohydrates (12); only seven sugars, including D-glucose (D-Glc), D-fructose (D-Fru), and D-galactose (D-Gal), are abundant in nature (13). Other monosaccharides, such as D-tagatose (D-Tag), D-allulose (D-Alu), D-allose (D-All), and D-sorbose (D-Sor), are referred to as rare sugars (14).

D-Glc derived from the Calvin cycle in autotrophic cells is a primary carbon source for living organisms. Hence, eukaryotes and prokaryotes have a substrate preference for D-Glc. For example, upon the decomposition of starch in the digestive tract, D-Glc is absorbed into the blood via sodium-glucose transport protein 1 (SGLT1), a glucose transporter in the small intestine (15). Due to the lack of a specific sugar transporter, the absorption rate of rare sugars, such as D-Tag (an isomer of D-Gal) and D-Alu (a 3-epimer of D-Fru), is low in the small intestine of animals, and calorigenic energy is also low (16). It is now clear that host diet shapes the composition of the gut microbiota (17, 18). Intriguingly, enteric gut microbes harmful to the host are adapted to diverse sugars, whereas health-promoting gut microbes show specific preferences toward rare sugars (19). Enzymatic systems for the synthesis and utilization of rare sugars have implications for the development of novel pharmaceuticals and/or drugs with antidiabetic, antihyperglycemic, and antitumor effects (14, 20).

We hypothesized that the inability to utilize rare sugars can be explained by the lack of accessibility to these sugars in nature during evolution. There are many unresolved issues regarding the evolution of sugar preferences, including how sugars define cellular sugar preferences (21), which genetic elements confer the gain of substrate preferences for available sugars, and whether preferences for abundant sugars can be altered when rare sugars become abundant. In this study, we tracked the molecular changes during bacterial cell culture in an environment enriched in a rare sugar, demonstrating the evolutionary trajectory governing adaptation for sugar preferences. We also redesigned the novel catabolic pathway in an engineered bacterium to validate the identified D-Tag utilization pathway. This engineered rare-sugar-utilizing strain provides a feasible cell-based screening system with applications for engineering non-phosphorylated sugar isomerases.

## RESULTS

**Heterologous expression of a class II aldolase initiated adaptive evolution for D-tagatose utilization in *Escherichia coli*.** Several pathogenic enteric bacteria and *Bacillus licheniformis* that can grow on D-Tag possess a D-Tag-specific PTS (e.g., *tagKTH*), including a class II aldolase (e.g., *gatY*) (Fig. 1A). Unlike abundant sugars [e.g., D-Glc, D-Fru, and L-arabinose (L-Ara)], the naturally occurring rare sugar D-Tag did not support the growth of *E. coli* BL21(DE3) (Fig. 1B). The inability of *E. coli* BL21(DE3) to utilize D-Tag could be explained by the lack of D-Tag-1P kinase and transporter subunits IIBC and IIA/HPr. Despite low sequence identity between *E. coli* fructose-specific transporters FruA (IIBC) and FruB (IIA), *K. pneumoniae* and *B. licheniformis* D-Tag transporter IIBC (34 to 36%), and *K. pneumoniae* D-Tag transporter IIA (34%) (Fig. 1A), the construction of gene arrays and clusters suggests that these endogenous sugar transporters may be involved in D-Tag uptake into cells due to their substrate promiscuity under specific selective pressure (22, 23). D-Tag-1,6-bisphosphate aldolase (D-Tag-1,6-BP aldolase) (GatY) is coregulated with the D-Tag PTS, including TagTH (transporter) and TagK (kinase) (24, 25). However, *E. coli* aldolases (i.e., GatY and KbaY) located in the galactitol (Gat) PTS or the *N*-acetyl-galactosamine/D-galactosamine (GalNAc/GalN) PTS are regulated by their corresponding substrates (26, 27). We thus hypothesized that aldol cleavage of phosphorylated D-Tag to dihydroxyacetone phosphate (DHAP) and glyceraldehyde-3-phosphate (GAP) could lead to metabolic flux for D-Tag utilization (Fig. 1C).

First, we overexpressed genes encoding *E. coli* class II ketose-bisphosphate aldolases, such as *E. coli* D-Tag-1,6-BP aldolases (EC_GatY and EC_KbaY) and *E. coli* D-Fru-1,6-BP

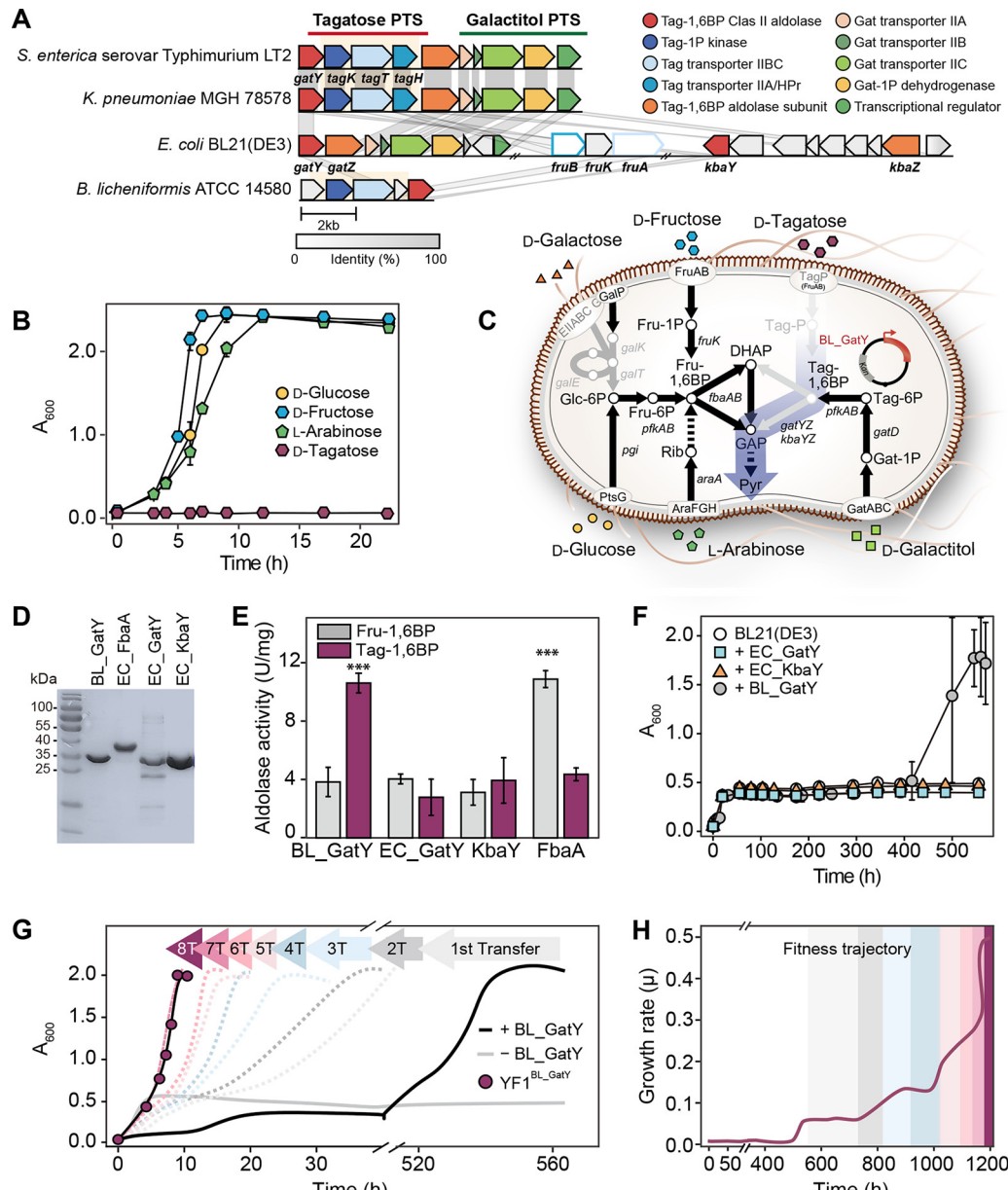

**FIG 1** Comparison of the composition of the D-tagatose catabolism-related gene clusters and design of the ALE experiment for conferring D-Tag usability. (A) Comparative genomics of *E. coli* BL21(DE3), enteric bacteria, and *Bacillus licheniformis* with the D-Tag phosphotransferase system (PTS) (24, 37, 62) and D-galactitol PTS by the Clinker (version 0.0.23). These D-Tag-PTS genes were heterologously expressed in *E. coli* and functionally characterized (24, 37). Heterologous expression conferred on *E. coli* the ability to utilize D-Tag following directed evolution (62). The cut-off identity was 30%; *E. coli* endogenous *fruB* and *fruA* showed 34% and 36% identities to enteric bacterial D-Tag transporter subunit IIABC and to transporter subunit IIBC of *B. licheniformis*. (B) Growth profiles of *E. coli* BL21(DE3) in M9 medium supplemented with 0.5% D-Glc, 0.5% D-Fru, 0.5% L-Ara, 0.5% D-Gal, or 0.5% D-Tag. (C) Scheme for novel rare sugar catabolic flux induced by the retroaldol reaction. Monosaccharide catabolic pathways of *E. coli* BL21(DE3) lack D-Gal and D-Tag catabolic machineries. (D) SDS-PAGE analysis of purified class II aldolases: BL_GatY, 30 kDa; EC_FbaA, 39 kDa; EC_GatY, 30 kDa; and EC_KbaY, 31 kDa. (E) Class II aldolase activity assay with two types of ketohexose 1,6-bisphosphate with D-Fru-1,6-BP and D-Tag-1,6-BP substrates. Comparisons were performed by analysis of variance (ANOVA) (***, $P < 0.001$, $n = 3$). (F) Phenotypic changes in *E. coli* BL21(DE3) harboring aldolases from *E. coli* and *B. licheniformis* in M9 medium supplemented with 0.05% D-Fru and 0.45% D-Tag. (G) Growth profiles of serially passaged ALE-induced strains. A strain lacking *BL gatY* did not evolve within 600 h (gray line). *E. coli* harboring *BL_gatY* evolved within 500 h (black line). Subcultured growth profiles of evolved strain with the BL_GatY plasmid by serial transfer, 1T to 8T, dotted lines; 8th transferred strain, YF1[BL_GatY]. (H) Fitness trajectory based on the growth rates of YF1[BL_GatY] strain in M9 medium supplemented with 0.05% D-Fru and 0.45% D-Tag. Error bars show standard deviations.

aldolase (EC_FbaA), and *B. licheniformis* class II D-Tag-1,6-BP aldolase (BL_GatY) in *E. coli* BL21(DE3) (Fig. 1D). After purification of the recombinant aldolases by His-tag affinity column chromatography, we assayed their aldol cleavage activity for D-Tag-1,6 bisphosphate (D-Tag-1,6-BP) (Fig. 1E). EC_GatY and EC_KbaY showed much lower D-Tag-1,6-BP cleavage activity than BL_GatY due to the lack of *gatZ* and *kbaZ* subunits, respectively, which are required for full aldolase activity, as reported previously for class II tagatose-1,6-BP aldolases, but not for *B. licheniformis* GatY ($\sim$30 kDa) (28). Accordingly, for heterologous expression in *E. coli* BL21(DE3), we chose the monomeric BL_GatY as the simplest protein complex with the highest aldol cleavage activity, which might reduce the metabolic burden and minimize the occurrence of protein aggregation and/or misfolded protein accumulation during long evolution periods. Next, we examined whether the enzymatic retroaldol (cleavage) reaction via BL_GatY, EC_GatY, or EC_KbaY confers D-Tag metabolism. To achieve this, we grew *E. coli* BL21(DE3) strains overexpressing three types of D-Tag-1,6-BP aldolases in M9 medium supplemented with 0.05% (wt/vol) D-Fru and 0.45% (wt/vol) D-Tag (Fig. 1F and Fig. S1 in the supplemental material). We employed D-Fru as a minimal helper substrate, which we expected would not only maintain a diverse gene pool due to an increase in cell density and improve cell viability in D-Tag-containing medium but also induce the expression of the D-Fru PTS. Upon the depletion of 0.05% D-Fru, wild-type (WT) *E. coli* cells harboring expression vectors for two endogenous aldolases, EC_GatY (ACT43848.1) and EC_KbaY (ACT44808.1), did not grow on D-Tag. However, the retroaldol reaction induced by the expression of the *BL_gatY* gene conferred growth on D-Tag after 500 h of incubation at 37°C. These results indicated that we could leverage BL_GatY as a driving force for large jumps in the fitness landscape.

Next, serial transfer of bacterial cultures into fresh M9 medium supplemented with D-Tag revealed the short-term evolutionary path (Fig. 1G). The first strains with BL_GatY grown on D-Tag in 500 h of incubation at 37°C went through several rounds of serial transfer with each population in the late-exponential phase until the 8th passage, when the highest growth rate of 0.5 ($\mu$) on D-Tag was reached (Fig. 1H). We chose cells at the 8th passage as an endpoint strain, YF1[BL_GatY]. BL_GatY promoted survival and adaptation in the presence of an enriched rare sugar; endogenous FruB (ACT43921.1) and FruA (ACT43919.1), which show >30% amino acid sequence identities to enteric bacterial counterparts, might function as a D-Tag PTS EIIA/HPr and an EIIBC, respectively (Fig. 1A and C). Subsequently, we removed the BL_GatY plasmid from the adapted YF1[BL_GatY] strain to examine whether the ability to metabolize D-Tag could be retained by endogenous genes only. The resulting YF1 strain without the BL_GatY plasmid could grow on D-Tag in about 20 h; however, an additional transfer on D-Tag readily yielded the YF2 strain, showing full growth with a rate similar to that of YF1[BL_GatY] without a lag time (Fig. 2A), suggesting that unidentified adaptive mutation(s) responsible for shortening the lag time occurred.

Whole-genome sequencing revealed five mutations in the YF1[BL_GatY] strain and six mutations in the YF2 strain compared to the *E. coli* BL21(DE3) WT strain (Fig. 2B and Table S1). We found three common mutation sites directly relevant to glycolysis in these adaptive laboratory evolution (ALE)-induced strains: *fruK* 115G→T, encoding the Ala39Ser mutation in 1-phosphofructose kinase (FruK_A39S); a 67-bp deletion in the catabolite repressor/activator (Cra, also known as FruR) binding site (CraBS_Δ67) (29, 30) of the 5′ untranslated region (UTR) of the *fru* operon, which depresses the expression of the *fruBKA* operon; and a substitution in the *lac* promoter region of the gene encoding T7 DNA-directed RNA polymerase (T7RNAP[Pro]) in the *E. coli* BL21(DE3) strain, regulating the expression level of the pET system (31–33). These three mutations were likely the minimal set for D-Tag utilization, because the only genetic difference between YF1[BL_GatY] and YF2 was a frameshift insertion-deletion (indel) mutation (a 17-bp deletion) introducing a premature termination codon (PTC) in the *N*-acetyl-galactosamine (GalNAc) repressor *agaR* (*agaR*_Δ17), encoding AgaR[PTC] of the YF2 strain (Fig. 2B). Differential growth profiles of the mutants indicated impaired Cra regulation of the *fruBKA* operon, suggesting that FruB, FruK, and FruA could be responsible for

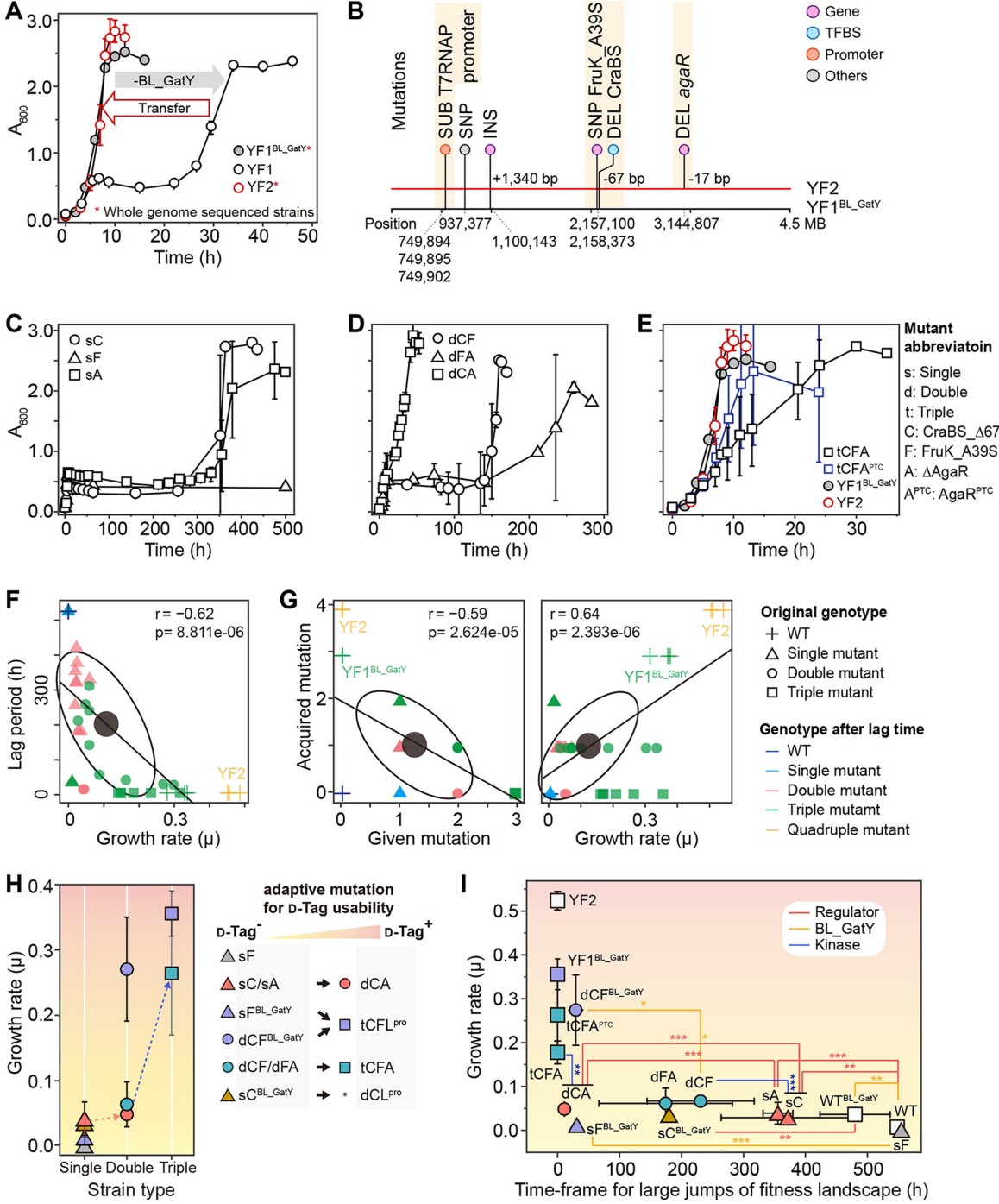

**FIG 2** Analysis of minimum genetic factors to utilize D-Tag. (A) Generation of optimized strains with and without the BL_GatY aldolase plasmid in M9 medium supplemented with 0.05% D-Fru and 0.45% D-Tag. YF1[BL_GatY] strain; YF1, BL_GatY plasmid-cured YF1[BL_GatY] strain; YF2, the subcultured strain of YF1 with growth rate similar to that of YF1[BL_GatY]. (B) Genotypic properties of evolved strains YF1[BL_GatY] and YF2. (C to E) Growth profiles of three types of mutant strains in medium with 0.05% D-Fru and 0.45% D-Tag: (C) single mutant strains (sC, CraBS_Δ67; sF, FruK_A39S; sA, ΔagaR), (D) double mutant strains (dCF, CraBS_Δ67 and FruK_A39S; dFA, FruK_A39S and ΔagaR), and (E) triple mutant strains (tCFA, CraBS_Δ67, FruK_A39S, and ΔagaR; tCFA[PTC], CraBS_Δ67, FruK_A39S, and AgaR[PTC]). (F) Pearson correlation analysis of relationships between the growth rates and lag periods for D-Tag usability in ALE-induced mutant strains. (G) Pearson correlation analysis of relationships between the numbers of given mutations and acquired mutations and between growth rates on D-Tag and the number of acquired mutations. (H) Analysis of the effects of mutant strain types on growth rates after ALE experiments. Tracking mutant strains leading to D-Tag usability (red dotted line) and determining optimized suboptimal growth (blue dotted line). L[pro], mutation of T7RNAP[Pro]. (I) Analysis of mutation effects on the acquisition of novel phenotypes requiring time for large jumps in the fitness landscape. Statistical significance was determined by *t*-test of lag periods (h) and growth rates (μ). ***, $P < 0.001$; **, $P < 0.01$; *, $P < 0.05$. Error bars show standard deviations.

the ability of the mutant to use D-Tag for growth. In addition, the FruK mutant might contribute to the phosphorylation of D-Tag for growth. Substitutions in T7RNAP$^{Pro}$ might control the expression level of the *BL_GatY* plasmid, because the *lacUV5* promoter was reverted to its original sequence, thereby reducing the expression level of T7 polymerase. Accordingly, the expression of BL_GatY might be critical in accelerating the adaptive evolution of catabolic pathways toward D-Tag utilization. Therefore, the mutations in ALE strains were prerequisites or crucial for D-Tag utilization. Furthermore, these stepwise changes support the notion that the retroaldol reaction drove ALE in *E. coli* grown on D-Tag.

**Rare sugar utilization required the loss of tightly regulated sugar-specific phosphotransferase systems.** To assess the effects of ALE-induced genome mutations on bacterial growth, we introduced the CraBS_Δ67, FruK_A39S, Δ*agaR*, and AgaR$^{PTC}$ mutations into WT *E. coli* BL21(DE3) to yield single (s), double (d), and triple (t) mutant *E. coli* BL21(DE3) strains (Table S2) and compared the growth rates and lag times between mutant strains on D-Glc (Fig. S2A to C), D-Fru (Fig. S2D to F), and D-Tag (Fig. 2C to E). All three single mutants grew well on D-Glc and D-Fru without any lag time (Fig. S2A and D) but showed a markedly extended lag time on D-Tag (Fig. 2C). Remarkably, sC (CraBS_Δ67) and sA (Δ*agaR*) strains with single mutations in regulatory genes could grow after a reasonable time frame, regardless of the presence or absence of BL_GatY, indicating that modulation of the sugar PTS regulatory system may confer the ability to utilize D-Tag via endogenous genes only. However, *E. coli* BL21(DE3) WT and sF (FruK_A39S) strains did not grow on D-Tag even after 500 h of incubation in the absence of BL_GatY. Accordingly, the expression of BL_GatY might be critical in accelerating the adaptive evolution of catabolic pathways toward D-Tag utilization.

In the case of double mutant strains, the dCF strain (CraBS_Δ67 and FruK_A39S, directly involved in sugar PTS) on D-Glc and D-Fru showed slower growth rates than the other two mutants, dFA (FruK_A39S/Δ*agaR*) and dCA (CraBS_Δ67/FruK_A39S) (Fig. S2B and E). In D-Glc and D-Fru-containing medium, the growth rates of various mutant strains decreased as the number of mutations increased (Fig. S2A to F). Thus, these results indicated that either FruK_A39S or CraBS_Δ67 influenced D-Fru utilization. These double mutants grew on D-Tag with a lag of ~150 h after depletion of 0.05% D-Fru (Fig. 2D). In D-Glc and D-Fru-containing medium, the growth rates of various mutant strains decreased as the number of mutations increased. dCA showed a short lag time; however, both dCF and dFA strains on D-Tag exhibited markedly extended lag times. On the other hand, the introduction of Δ*agaR* eliminated the lag phase in the mutant strains, likely due to the perturbation of AgaR repression of the *kbaZ-agaVWEFA* (NC_012971.2, bases 3145107 to 3150123) and *agaS-kbaY-agaBCDI* (NC_012971.2, bases 3150473 to 3155524) operons, increasing the expression of the aldolase KbaY [26]. Indeed, the triple mutant tCFA and tCFA$^{PTC}$ strains grew without a lag period, except for slower growth than the YF1$^{BL\_GatY}$ and YF2 strains (Fig. 2E). Intriguingly, tCFA and tCFA$^{PTC}$ showed little lag for bacterial growth on D-Glc; however, they showed a long lag period on D-Fru (Fig. S2F). Therefore, three mutations (CraBS_Δ67, FruK_A39S, and Δ*agaR* [or AgaR$^{PTC}$]) were sufficient to support bacterial growth on D-Tag, suggesting that gain of D-Tag preference is due to a trade-off with D-Fru specificity (Fig. S2E and F).

We further analyzed correlations between genotypic and phenotypic changes to identify the minimal genetic elements for rare sugar utilization. For various mutant strains with an ability to use D-Tag (Tag$^{+}$), we detected linear correlations (based on Pearson correlation coefficients, *r*) between four kinds of phenotypes (i.e., growth rate, lag period, and given and acquired mutations) (Fig. S3). A shorter lag period on D-Tag showed a negative correlation with the number of mutations in the starting strain (Fig. 2F and Fig. S3). We also found that regardless of the number of mutations, all mutants with increased growth rates on D-Tag shared four types of mutations (i.e., CraBS_Δ67, FruK_A39S, Δ*agaR* [or AgaR$^{PTC}$], and T7RNAP$^{Pro}$), suggesting that only these four mutations are required to obtain an ability to use D-Tag (Fig. 2G). Indeed, single and double mutant strains with impaired regulation in sugar utilization begun to grow or grew more rapidly on D-Tag with additional regulation-related mutation(s) toward the same genetic features (Fig. 2H). Therefore, we concluded that

mutations in regulators of PTS expression, such as CraBS_Δ67 and AgaR[PTC], could result in the gain of an ability to use D-Tag.

Notably, BL_GatY-harboring strains (e.g., sF[BL_GatY], sC[BL_GatY], and dCF[BL_GatY]) with the Tag[+] phenotype accompanied by substitutions in the T7RNAP[Pro] indicated that the expression level of BL_GatY is likely critical for D-Tag utilization. The transcription level of the *lacUV5* (−35, TTTACA; −10, TATAAT) promoter of the gene encoding T7RNAP should be reduced to that of the original *lac* promoter sequence (−35, TTTACA; −10, TATGTT) to minimize the metabolic burden of cells harboring the pET-28a(+)-BL_GatY plasmid on D-Tag (33), resulting in an increase in bacterial growth rates (Fig. 2I). It was thus determined that two minimal mutations (i.e., dCA) conferred the ability of *E. coli* to utilize D-Tag, and additional mutations (i.e., tCFA or tCFA[PTC]) yielded optimized strains capable of utilizing D-Tag efficiently. The optimized strains did not have a lag time on D-Tag or D-Glc (Fig. S4A), but they did on D-Fru, irrespective of the presence or absence of the BL_GatY plasmid (Fig. S2F and S4B). We found that either the deletion in *agaR* for YF2 or mutations in the T7RNAP[Pro] for YF1[BL_GatY] conferred growth on D-Tag. Therefore, the retroaldol reaction is crucial for obtaining the ability to use new sugar sources. Taken together, the retroaldol reaction induced by either BL_GatY- or AgaR-mediated endogenous aldolase(s) contributed to an ability to utilize D-Tag by reducing the lag period, and appropriate expression levels of aldolases accelerated the growth rates (Fig. 2I). The introduction of the FruK_A39S mutation to strains with mutations affecting regulation of the Fru PTS and AgaR-controlled aldolase further increased growth rates.

Nevertheless, the combination of mutations indicated that PTS activity and the expression levels of individual components contribute differentially to the evolution of rare sugar utilization. A differential-expression analysis of the WT and CraBS_Δ67-containing mutant strains (i.e., dCF and YF2) showed two noticeable features relevant to the catabolic pathway for D-Tag. In the case of dCF and YF2 strains, the expression level of the *fru* operon comprising the transporter and kinase was >2-fold higher than that in the WT strain on all types of sugars examined, especially on D-Glc (Fig. 3A and Table S4). We also found that *kbaY* was upregulated in YF2 on all sugars and in dCF without GatY on D-Tag, irrespective of the presence or absence of 0.05% D-Fru, suggesting that KbaY acts as a D-Tag-1,6-BP aldolase, instead of BL_GatY (28). In addition, the expression of T7RNAP in YF2 and YF1[BL_GatY] strains was lower than that in the dCF strain on D-Tag, indicating that mutations in the *lacUV5* promoter in the WT and dCF strains reduced the expression of T7RNAP (Fig. 3A). These phenotypes accounted for the key roles of FruBKA in transporting and phosphorylating D-Tag and KbaY to catalyze the aldol cleavage of D-Tag under the loss of Cra repressor binding by the deletion in CraBS (Fig. S5A and B). Therefore, we concluded that the CraBS deletion interrupts the tight regulation of the *fru* operon encoding FruBA for the transport of D-Tag and FruK_A39S for phosphorylation to yield D-Tag-1,6-BP, followed by cleavage by KbaYZ via the AgaR[PTC] mutation.

To further validate the novel D-Tag catabolic pathway, we constructed single knockout strains for the transporter, kinase, and aldolase homologs, consisting of D-Fru PTS and D-galactosamine (GalN) PTS, in the cured YF2 strain and confirmed the D-Tag utilization profile (Fig. 3B and Table S2). Deletion of *fruA* encoding the D-Fru transporter resulted in delayed growth in D-Fru, as expected, and no growth on D-Tag (Fig. 3B and Fig. S5D). Similarly, when FruK_A39S was deleted, we detected delayed growth in D-Glc and D-Fru medium and no growth on D-Tag (Fig. 3B and Fig. S5C and D). Notably, deletion of *kbaY* located in the GalN PTS in the Δ*kbaY*_YF2 strain eliminated D-Tag utilization (Fig. 3B). This aldolase shares very high sequence identity with *gatY* aldolases of *K. pneumoniae* (63%) and *B. licheniformis* (55%) (Fig. 1A). Deletion of CraBS induced sufficient expression of the *fru* operon, thereby phosphorylating D-Tag, which is plugged into the glycolysis pathway by KbaY aldolase (Fig. 3C). Based on these results, we predicted that deletion of the Cra binding site would prevent the inhibition of *fru* operon expression and sufficiently induce the expression of the enzymatic machinery for D-Tag utilization (30). We also found that the *agaR* deletion likely regulated the expression of the *kbaY* gene encoding D-Tag-1,6-BP aldolase in the *aga* operon to direct the metabolic flux to glycolysis. In short, the novel metabolic flux was the result of a retuning of gene

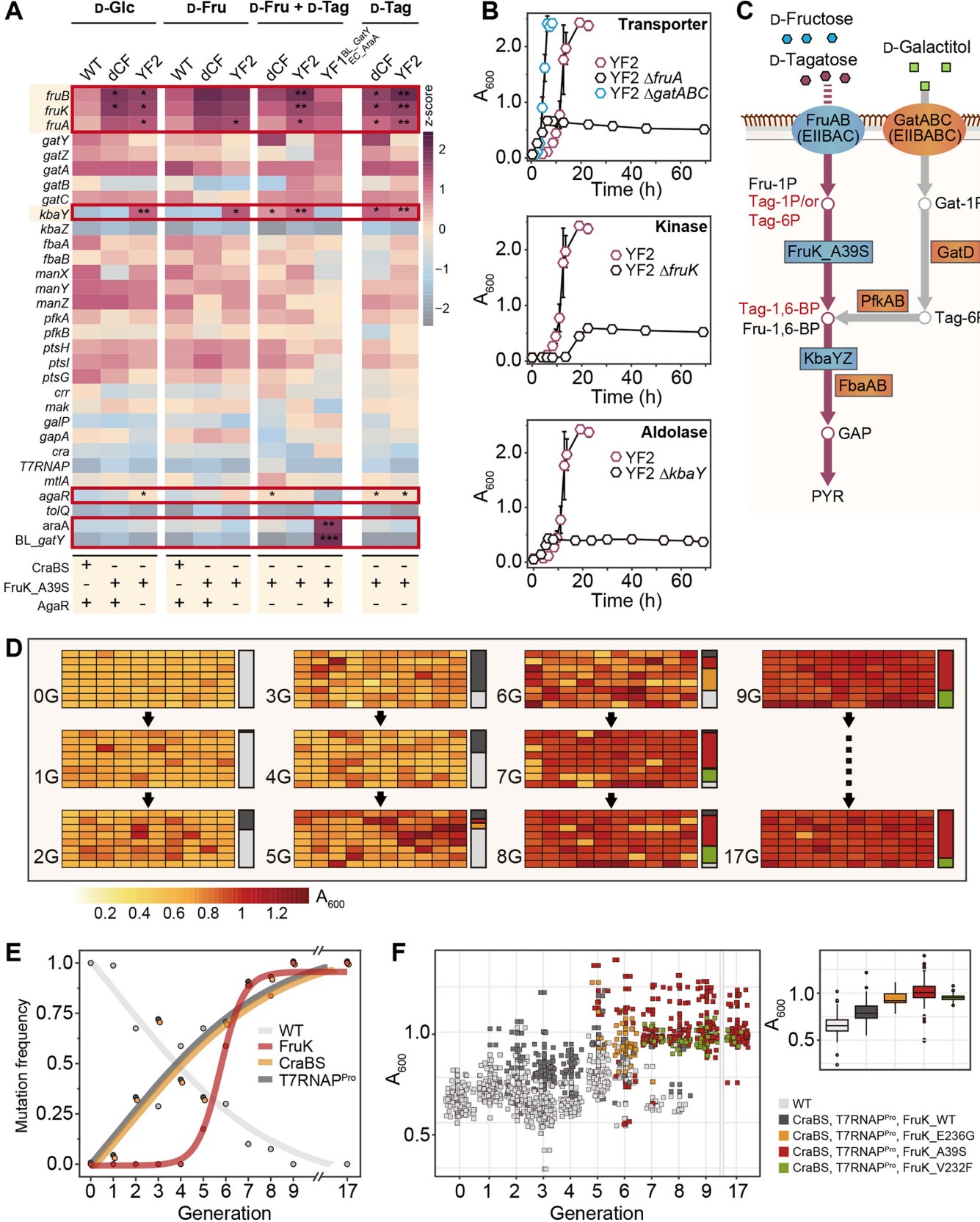

**FIG 3** Identification of a novel D-Tag catabolic pathway in *E. coli* and tracking the adaptive evolutionary trajectory of bacterial cells toward nonutilizable sugars. (A) Analysis of mRNA expression levels of sugar PTS-related genes with growth in four types of M9 sugar medium supplemented with 0.5% D-Glc, 0.5% D-Fru, 0.5% D-Tag, 0.05% D-Fru, and 0.45% D-Tag. Genotypes of cultured strains under each condition. The expression levels were normalized to those

expression pathways provoked by the deletion of key regulatory elements. Despite the low sequence identities of the *fruBKA* operon and authentic Tag transporters (IIBC and IIA) from *B. licheniformis* or *Klebsiella* and D-Tag-1P (or -6P) kinase from *Klebsiella*, the loss of a tight regulation of the *fru* operon allowed the D-Fru PTS to efficiently transport and phosphorylate D-Tag (Fig. 1A and 3C).

**Order-specific minimal genetic repertoires for acquiring rare sugar catabolic pathways.** We investigated the order of mutations involved in the acquisition of the ability to use D-Tag by tracking mutations in each subpopulation derived from serial transfers of *E. coli* YF1^BL_GatY cells grown on D-Tag (Fig. 3D). We randomly isolated 96 single colonies from the subcultured strains (Fig. 1G) in every generation (0G to 17G) and transferred these to 96-well plates containing M9 medium supplemented with 0.05% D-Fru and 0.45% D-Tag. These plates were incubated for 12 h at 37°C and subjected to DNA sequencing of three adaptive mutation sites (i.e., *fruK*, the Cra binding site, and the T7RNAP^Pro region) found in the D-Tag-utilizing YF1^BL_GatY strain (Fig. 3D). PTS regulation-related mutations were detected in the early stage of evolution, followed by sugar kinase mutations in later stages.

Noticeably, the WT population decreased during subculturing and completely disappeared in 9G (Fig. 3E). In contrast, the populations with mutations in CraBS and T7RNAP^Pro were first detected at 1G and became dominant at 4G. Remarkably, new populations with several mutations (e.g., A39S, V232F, or E236G) in FruK appeared from 5G (Fig. 3F). Although these FruK variants coexisted, E236G decreased from 7G, whereas the frequency of V232F decreased as the number of serial transfers increased. However, after 17G, A39S became dominant. Furthermore, over time, the average optical density at 600 nm (OD$_{600}$) value increased. These results demonstrate that the population with mutations in *fruK*, the Cra binding site, and the T7RNAP^Pro would be the most competitive in D-Tag medium based on the correlation between generations elapsed and bacterial growth rates (Fig. 3D to F). Collectively, these data demonstrate that when BL_GatY catalyzes the retroaldol cleavage of D-Tag, the D-Tag catabolic pathway is likely to evolve through mutation(s) in the regulatory region of D-Fru PTS, with subsequent fine-tuning of D-Tag flux via mutation(s) in FruK.

In conclusion, the evolutionary trajectory involves the control of *fru* operon expression that occurs due to the deletion of the Cra binding site in the early stages of evolution. This induces sufficient expression of the genetic components required for D-Tag utilization. Furthermore, D-Tag flux might be adjusted by cleaving D-Tag-1,6-BP at an appropriate rate by the T7RNAP^Pro mutation, which could control the expression of the foreign *gatY* cloned in the pET plasmid. Finally, substrate specificity would be shifted, with increases in the growth rate and rare sugar utilization by kinase mutations.

**Additional phosphosugar kinase mutations for adjusting metabolic sugar fluxes.** We examined whether FruK variants (i.e., A39S, V232F, and E236G) from 5G to 17G show altered preferences for phosphorylated D-Fru and D-Tag. For this, we generated single mutant FruK enzymes and assayed the kinase activities of purified enzymes for D-Fru-1P, D-Fru-6P, and D-Tag-6P (excluding D-Tag-1P due to synthesis failure). The kinase activities of both FruK_E263G and FruK_A39S for primary D-Fru-1P were much lower than that of FruK_WT (Fig. S6A to C). On the other hand, FruK_A39S showed relatively higher activities for D-Tag-6P than FruK_WT, which became susceptible to nonprimary substrates. CB-Dock-based molecular docking revealed that D-Fru-1P fitted well into the AlphaFold-aided three-dimensional (3-D) structural model of FruK_WT but not into that of FruK_A39S, presumably due to inappropriate coordination of the phosphoryl group of Fru-1P proximal to the phosphate-binding site (Fig. S6E). In contrast, FruK_A39S coordinated to the C-6 carbon of

**FIG 3** Legend (Continued)
of the *rpoD* gene. Statistical significance was calculated from a *t*-test of z-scores. \*\*, $P < 0.01$; \*, $P < 0.05$. (B) Phenotypic changes in single-knockout YF2 strains in medium with 0.05% D-Fru and 0.45% D-Tag to identify the D-Tag PTS in *E. coli*. (C) Novel D-Tag catabolic pathway in the ALE-induced *E. coli* YF2 strain. (D) Cross-generational phenotypic changes and cell densities ($A_{600}$) of serially passaged ALE-induced strains from 0G to 17G. Each bar, next to plate well depictions, indicates the relative abundances of *E. coli* WT and mutant strains. gray, *E. coli* WT; dark gray, *E. coli* CraBS_Δ67 and T7RNAP^Pro; orange, *E. coli* CraBS_Δ67, T7RNAP^Pro, and FruK_E236G; red, *E. coli* CraBS_Δ67, T7RNAP^Pro FruK_A39S; olive, *E. coli* ΔCraBS_Δ67, T7RNAP^Pro and FruK_V232F. (E) Mutation frequencies for three loci: *fruK*, CraBS, and T7RNAP^Pro. (F) Genetic changes and average cell densities. *E. coli* WT and four types of mutants: *E. coli* CraBS_Δ67 and T7RNAP^Pro; *E. coli* CraBS_Δ67, T7RNAP^Pro, and FruK_E236G; *E. coli* CraBS_Δ67, T7RNAP^Pro FruK_A39S; and *E. coli* ΔCraBS_Δ67, T7RNAP^Pro, and FruK_V232F. The panel on the right shows average OD ($A_{600}$) values of cell populations corresponding to *E. coli* WT and mutant strains. Error bars show standard deviations.

phosphorylated D-Tag (i.e., Tag-1P or Tag-6P) proximal to the mutation sites might contribute sufficient room for accepting ATP as a phosphodonor and an adequate orientation of the acceptor hydroxyl group of sugar (Fig. S6F to G), indicating that the FruK mutation renders the host strain able to access phosphorylated D-Tag as a new carbon source. The A39S mutation, predominantly found in 17G, the endpoint of the evolution experiment, might establish glycolytic flux with D-Tag. We concluded that the catabolic flux for a rare sugar would be redirected by mutations in regulators and then fine-tuned by mutations in sugar kinases, resulting in the evolution of a new catabolic pathway.

**A redesigned tagatose catabolic pathway enables galactose utilization via arabinose isomerase.** As a further validation of the newly evolved D-Tag catabolic pathway, we heterologously expressed L-arabinose isomerase (AI) as D-galactose isomerase (GI), responsible for the conversion of D-Gal to D-Tag (34), in *E. coli* BL21(DE3) YF1$^{BL\_GatY}$, which is incapable of D-Gal utilization due to the lack of the Leloir pathway (Fig. 4A and B). We complemented bacterial growth on D-Gal via the D-Tag pathway. The Δ*araA* YF1$^{BL\_GatY/EC\_AraA}$ strain could grow in D-Gal medium only in the presence of the *araA* plasmid because expressed AI converted accumulated intracellular D-Gal as a substrate for D-Tag (Fig. 4C and D). We detected mRNA expression patterns similar to that with D-Tag, indicating that the converted enzymatic product of D-Gal was catabolized by the new D-Tag pathway (Fig. 4E).

Next, we applied this ALE-induced *E. coli* strain favoring various sugars as a sugar isomerase screening platform. The workflow is summarized in Fig. 4A. The directed evolution of sugar isomerase in this system requires the construction of a platform host able to grow solely via AI activity. The ALE-induced *E. coli* BL21(DE3) D-Tag-utilizing strain meets this requirement. We deleted the chromosomal *araA* to prevent interference with endogenous AI activity. Therefore, our recombinant Δ*araA* YF1$^{BL\_GatY}$ could be used as a screening host to identify AIs with improved D-galactose isomerizing activity. This recombinant strain can provide a cell growth-associated screening system for engineering AI to GI.

**D-Tag-utilizing strains can serve as a platform for engineering AI.** To verify the capacity of the ALE-induced strain for directed evolution, we first generated an *araA* random mutagenesis library (>1 × 10$^6$ variants), which was cloned into a pET-22b(+) vector and transformed into *E. coli* YF1$^{BL\_GatY}$. The mutant library (ML) strains, YF1$^{BL\_GatY/EC\_AraA\ ML}$, were grown on D-Gal medium. Cell growth was initially observed at about 300 h of incubation, whereas after the second transfer in M9 medium supplemented with D-Gal, full growth was observed within 48 h (Fig. 4F). Additional rounds of transfer resulted in much faster growth of mutant libraries, with higher growth rates than those of the control strain harboring the WT AI plasmid. Serial transfers resulted in homogenous dominant populations until the maximal growth rate was reached. After the last round of transfer, we extracted plasmids from individual colonies, retransformed them into freshly prepared competent recombinant *E. coli* YF1$^{BL\_GatY}$ cells, and grew them for comparison with the control strain in D-Gal medium (Fig. 4G). Since AraA is a homohexameric enzyme, it is difficult to obtain positive mutants using random mutagenesis due to the very low yield (<0.1%) of positive clones (35, 36). As the doubling time of selected mutants is ~20 h (Fig. 4G), growth-based selection requires a long period of time to achieve colonization and/or dominance of a minor population of positive mutants (Fig. 4F). Consequently, we screened out three prominent strains that exhibited much faster growth than the control strain. A sequencing analysis of plasmids extracted from these strains revealed that H17R/R159S/V168A, E22D/M95L/H157L, and V368A/E493D mutations of EC_AraA were responsible for rapid-growth phenotypes on D-Gal.

To identify whether EC_AraA variants identified using *in vivo* screening systems can exhibit altered substrate affinity toward D-Gal, we characterized biophysically and biochemically purified AI variants (Fig. S7A and B). Variants (EC_AraA H17R/R159S/V168A, E22D/M95L/H157L, and V368A/E493D) had high levels of structural similarity to the WT enzyme (Fig. S7C) and exhibited an increased conversion yield of about 3.4% in 96 h, indicating improved conversion activity, probably due to the altered substrate affinity toward D-Gal over L-Ara (Fig. 4H). The WT enzyme exhibited a lower $K_m$ value, a higher turnover number ($k_{cat}$), and higher catalytic efficiency ($k_{cat}/K_m$) for L-Ara than for D-Gal,

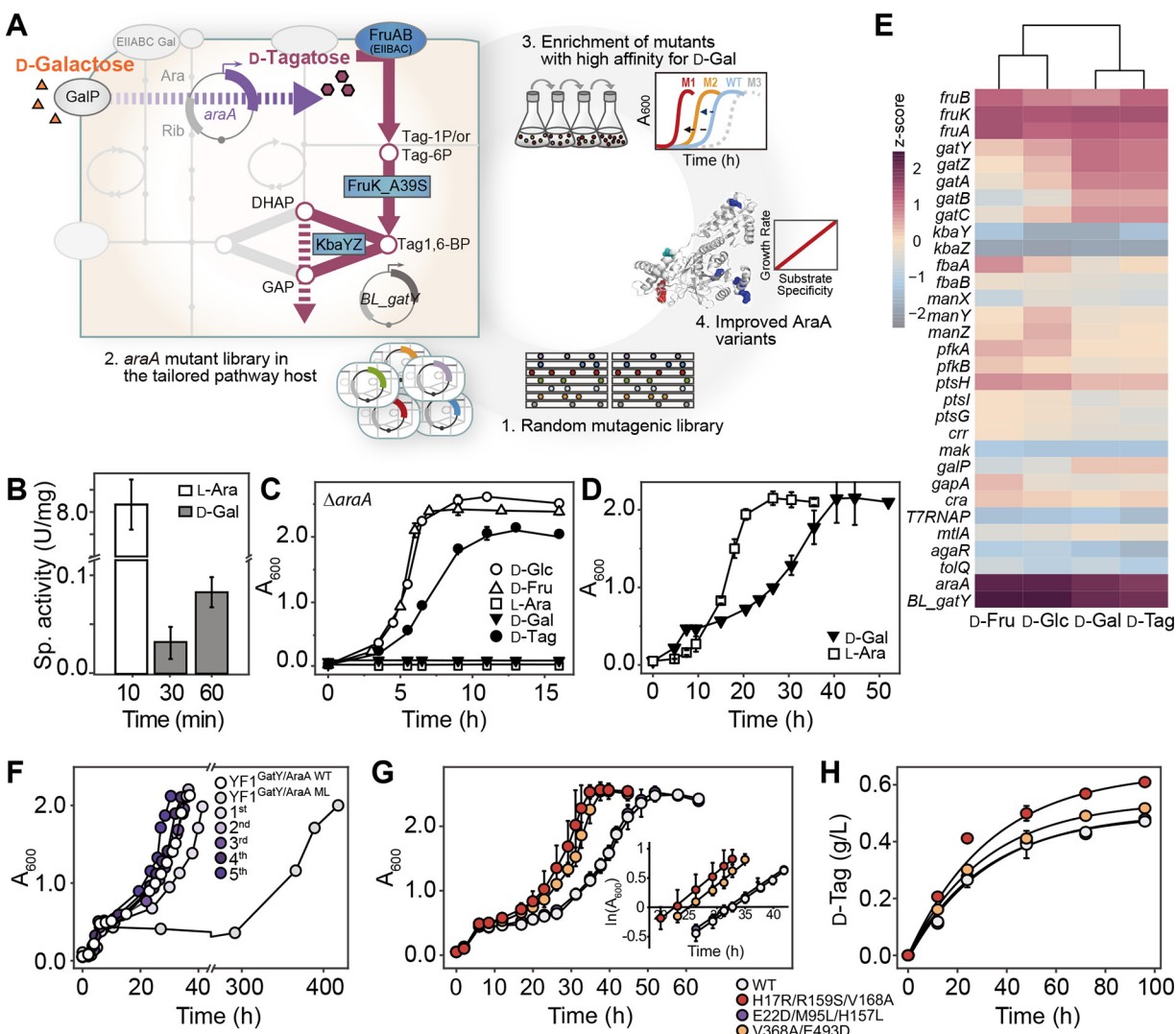

**FIG 4** Indirect D-Tag catabolic pathway starting from D-Gal. (A) Redesign of the *E. coli* sugar catabolic pathway, using D-Gal in an indirect D-Tag pathway by L-arabinose isomerase (AI). (B) Specific activities of AI with L-Ara and D-Gal as substrates. (C) ΔaraA YF1<sup>BL_GatY</sup> growth profiles in M9 medium supplemented with each of five types of 0.5% (wt/vol) sugars, D-Glc, D-Fru, L-Ara, D-Gal, and D-Tag. (D) Growth profiles of YF1<sup>BL_GatY/EC_AraA</sup> in M9 medium supplemented with 0.5% L-Ara or 0.05% D-Fru and 0.45% D-Gal. (E) Analysis of mRNA expression levels of sugar PTS-related genes of the YF1<sup>BL_GatY/EC_AraA</sup> strain in four types of sugar medium. The expression levels were normalized to those of the *rpoD* gene. (F) Growth profiles of *E. coli* BL21(DE3) harboring pET-22b(+)-EC_AraA ML random mutant libraries in M9 medium supplemented with 0.05% D-Fru and 0.45% D-Gal. (G) Growth profiles of ΔaraA YF1<sup>BL_GatY</sup> strains expressing EC_AraA variants (i.e., H17R/R159S/V168A, E22D/M95L/H157L, and V368A/E493D) and comparison of specific growth rates for AI mutant strains. (H) Time course of D-Tag production during AI-catalyzed isomerization. The experiments were performed in duplicate. Error bars show standard deviations.

consistent with previous observations (Table S5). Remarkably, the $K_m$ values for all three EC_AraA variants for D-Gal were approximately 2 to 3 times lower than those of the wild type, indicating that these variants had a much greater substrate preference for D-Gal. Accordingly, the newly developed *in vivo* screening system successfully selected improved variants with a higher affinity toward $C_6$ sugars.

## DISCUSSION

The diversity of sugar catabolic pathways, including sugar PTS, enables heterotrophs to efficiently consume abundant sugars (e.g., D-Glc and D-Fru) derived from autotrophs. The evolution of mechanisms to take advantage of rare sugars as primary carbon sources is expected to be relatively difficult. We evaluated how bacterial cells gain the ability to use nonusable sugars as a new carbon source. While *E. coli* was incapable of utilizing D-Tag as a rare sugar, a few pathogenic enteric bacteria could use it as a carbon source due to the

presence of a D-Tag-specific PTS (Fig. 1A) (24). Remarkably, however, D-Tag-utilizing *B. licheniformis* possessed the D-Fru PTS (25, 37), which is very similar to *E. coli* counterparts, except for GatY, responsible for the retroaldol cleavage of the $C_6$ sugar substrate. These results led us to examine whether implementation of the retroaldol reaction prompts adaptive evolution in which bacterial cells gain the ability to use D-Tag under excessive amounts of rare sugars. Under such unnatural conditions, we were able to track how heterotrophs acquire the ability to use a novel sugar and acquire the sugar-specific preference.

We showed that the availability of a rare sugar and appropriate aldolases for their phosphorylated sugars could drive enteric bacteria to use a nonusable sugar within 500 h, corresponding to ~500 generations in *E. coli* grown in M9 medium (Fig. 1F). The retuned descendant acquired the ability to use a rare sugar when we applied two drivers: a D-Tag-abundant environment and BL_GatY. Indeed, under both selective pressures, *E. coli* cells developed a new catabolic pathway for D-Tag flux. *E. coli* BL21(DE3) has its own D-Tag- 1,6-BP aldolases, GatY and KbaY, on each Gat and GalNAc PTS pathway. Both *E. coli* D-Tag- 1,6-BP aldolases cleave D-Tag- 1,6-BP produced from D-Tag-6-P by *pfkA* (28, 38, 39). Despite the presence of endogenous genes encoding phosphorylated D-Tag kinase, WT *E. coli* cannot use D-Tag, implying either a lack of D-Tag-specific PTS or strong repression of related genes. Unlike other GatY homologs, which must be assembled with GatZ, monomeric BL_GatY with sufficient aldolase activity could enable survival and adaptive evolution in the presence of an enriched rare sugar, indicating that BL_GatY is a convenient and efficient genetic target for evolutionary experiments.

The genotyping of individual bacterial cells derived from populations corresponding to the WT progenitor and adaptive variants during serial transfer allowed us to track the evolutionary trajectory of traits (Fig. 1 and 3). Resequencing integrated with quantitative real-time PCR revealed the adaptive mutations involved in altering the glycolytic reactions and the PTS. In a holistic view of how cells harness novel monosaccharides, we assumed that variation in glycolytic protein units, native genes encoding sugar PTS components, and regulatory factors play crucial roles in unlocking novel metabolic fluxes (Fig. 1C and 3C). Glycolytic enzymes are highly conserved across broad taxa (40), with multiple substrates per enzyme in addition to a primary substrate (41). In the PTS, most transporters tend to have low-level activity toward various monosaccharides. D-Fru, widely available in nature, is transferred by various membrane-spanning transporters, including *fruAB*, as well as *manXYZ*, *mtlA*, and *gutA*, with differences in primary substrate specificity. Loss of the Cra (catabolite repressor/activator) binding sites upstream from the *fru* operon during adaptive evolution is advantageous compared to loss of the *cra* gene itself, because it allows adaptation of the FruPTS to use a new ketohexose, which does not act as an inducer. We predicted that glycolytic enzymes have promiscuous activity (42, 43) toward various substrates to improve cell survival, providing machinery for different carbohydrates and a basis for adaptive evolution. We proposed that the gain of appropriate aldolases, followed by marginal adaptive mutations in endogenous kinases, can initiate the utilization of a rare sugar and that adaptive mutations in the 5′ UTR of a PTS with promiscuous characteristics can induce a retuned sugar flux. In addition, we anticipated that an appropriate aldolase expression level would be important because *lacUV5* of *E. coli* BL21(DE3) was mutated back to its original sequence, resulting in a reduction in the T7RNAP expression level. We examined whether a relatively weak promoter resulted in a shorter lag in bacterial growth on D-Tag than a strong T7*lac* promoter (data not shown). Indeed, the *E. coli* K-12 BW25113 strain harboring the pBAD-BL_GatY plasmid grew on D-Tag after ~200 h, which is a much shorter lag time than that of the *E. coli* BL21(DE3) strain (data not shown). Nevertheless, the *E. coli* BL21(DE3) strain lacking *galETK* genes can serve as a cell-based screening host for AraA-aided D-Tag production from D-Gal.

Finally, we examined whether an AI-aided isomerization of D-Gal to D-Tag can coordinate the D-Tag catabolic pathway comprising FruK_A39S and BL_GatY in *E. coli* BL21 (DE3), which cannot grow on D-Gal (Fig. 4A and C). Indeed, heterologous expression of AI responsible for the isomerization of D-Gal to D-Tag supported bacterial growth via the disclosed D-Tag pathway in the engineered *E. coli* BL21(DE3). The use of the D-Tag catabolic

pathway in this strain was validated by quantitative real-time PCR (qRT-PCR) (Fig. 4E), demonstrating the potential application of the rare sugar-utilizing strain as an *in vivo* high-throughput screening system for tailor-made sugar isomerases. According to the Izumoring strategy (44), all naturally occurring rare sugars with nutraceutical and pharmaceutical value can be produced from low-cost, abundant sugars (e.g., D-Glc, D-Fru, and D-Gal) via novel sugar isomerases. However, in meeting industrial demand, the screening of novel sugar isomerases and directed evolution of enzymes hinder the development of a biological process for the production of novel rare sugars. Conventional organic solvent-based isomerization assays cannot resolve the number of carbon atoms in sugars and their stereo configuration and are limited for high-throughput screening due to the complex assay procedures and substantial variation. In this regard, a cell-based assay system for the directed evolution of sugar isomerases would be efficient, non-labor-intensive, and cost-effective, as demonstrated by the AI-dependent growth on nonutilizable D-Gal in the ALE-induced D-Tag auxotrophic strain (Fig. 4C).

Among AI mutant libraries (up to $10^8$ unique variants) generated by random mutagenesis, several variants with higher isomerization activity of D-Gal to D-Tag supported the faster growth of the host strain on D-Gal in an enzyme activity-dependent manner. Five rounds of serial transfer on D-Gal medium resulted in the expression host with the most active AI mutant exhibiting faster growth than the other expression hosts with parental and negative mutant AIs (Fig. 4F). Consequently, we obtained an AraA variant with the highest activity (i.e., H17R/R159S/V168A) in this study. Previously, several attempts have been made to improve the D-Gal isomerizing activity of AraA (45–47). Among them, the AraA H18T variant from *Geobacillus stearothermophilus* (46) is comparable to H17R in our study; both mutations contribute to increasing the accessibility to D-Gal bound in the binding pocket of AraA.

This result indicates that our system is a powerful tool for the directed evolution of sugar isomerases with low-calorigenic rare sugars (e.g., D-Tag, D-Alu, D-All, and D-Sor) showing antidiabetic, antihyperglycemic, and antitumor activities, making these attractive alternative additives in the food and pharmaceutical industries.

## MATERIALS AND METHODS

**Bioinformatics analyses.** A synteny analysis of the whole-genome alignments to the D-Tag/D-Fru and galactitol PTS gene clusters was performed using the following microbial genome sequences obtained from GenBank: *Escherichia coli* strain BL21(DE3) (GenBank accession number GCF_000022665 .1), *Salmonella enterica* serovar Typhimurium strain LT2 (GCF_000006945.2), *S. enterica* serovar Typhi strain Ty2 (GCF_000007545.1), *Klebsiella pneumoniae* strain MGH 78578 (GCF_000016305.1), and *Bacillus licheniformis* strain ATCC 14580 (GCF_000011645.1). Sequences were clustered with a cut-off identity and visualized using Clinker version 0.0.23 (48).

**Bacterial strains and culture conditions.** The *E. coli* strains, plasmids, and primers used in this study are listed in Tables S2 to S4. Cultures were grown in Luria-Bertani (LB) medium and minimal medium (M9) at 37°C with kanamycin (Kan) at 50 $\mu$g mL$^{-1}$ or ampicillin (Amp) at 100 $\mu$g mL$^{-1}$. *E. coli* DH5$\alpha$ was used as a bacterial host for cloning, and *E. coli* BL21(DE3) was used for protein expression. For carbon utilization study, *E. coli* strains were grown aerobically in minimal medium (M9) supplemented with 0.5% (wt/vol) of each carbon source (e.g., D-Glc, D-Fru, D-Gal, L-Ara, and D-Tag; Sigma, St. Louis, MO, USA) at 37°C, harvested by centrifugation at 1,500 $\times$ g for 20 min, washed twice with M9 medium without a carbon source, and resuspended in appropriate fresh medium. Inoculated cultures (OD$_{600}$ of 0.05) were grown aerobically at 37°C, and bacterial growth was monitored by measuring absorbance at 600 nm (OD$_{600}$) with an Ultraspec 8000 spectrophotometer (GE Healthcare, Piscataway, NJ, USA). Unless otherwise stated, bacterial culture experiments were performed in triplicate.

**ALE.** In the process of adaptive laboratory evolution (ALE), recombinant *E. coli* strains harboring each aldolase-coding plasmid were grown on M9 medium supplemented with 0.05% (wt/vol) D-Fru, 0.45% (wt/vol) D-Tag, 0.2 mM IPTG (isopropyl-$\beta$-D-thiogalactopyranoside), and 50 $\mu$g mL$^{-1}$ Kan in a total volume of 25 mL at 37°C until cell growth was observed. Grown cells harboring pET-28a (+)-BL_GatY were serially transferred in M9 medium under the same conditions. The BL_gatY plasmid in the strain at the 8th passage (9G, YF1$^{BL\_GatY}$) was cured by high-voltage electroporation (49). Cured YF1 strains were grown at 37°C in M9 medium supplemented with 0.05% (wt/vol) D-Fru, 0.45% (wt/vol) D-Tag, without antibiotics, and with IPTG as an inducer until the growth rate reached that of the YF1$^{BL\_GatY}$ strain.

**Aldolase and kinase cloning, expression, and purification.** The genes encoding aldolases (e.g., *BL_gatY*, *EC_gatY*, *EC_kbaY*, and *EC_fbaA*) and kinases (EC_ *fruK*_WT, 115G$\rightarrow$T [FruK_A39S], 694G$\rightarrow$T [FruK_V232F], and 710A$\rightarrow$G [FruK_E236G] genes) were amplified by PCR using genomic DNA from *B. licheniformis* strain 14580 and *E. coli* BL21(DE3). The PCR mixture (50 $\mu$L) contained 20 ng genomic DNA,

10 pmol each primer (Table S3), 1 × PCR buffer, 0.2 mM deoxynucleoside triphosphate (dNTP) mix, and 2.5 U of PrimeSTAR HS DNA polymerase (TaKaRa Co., Shiga, Japan). The PCR product was cloned into pTOP Blunt V2, and the resultant construct was transformed into *E. coli* DH5$\alpha$ competent cells. Transformants containing pTOP Blunt V2 (Enzynomics, Daejeon, South Korea) harboring the gene encoding aldolase were selected on LB-Kan plates. Plasmid DNA was isolated from the transformants with inserts and digested with NdeI and XhoI. The digested DNA was purified and ligated into the NdeI and XhoI sites of pET-28a(+) (Novagen, Madison, WI, USA).

For the overexpression and purification of recombinant plasmids encoding class II ketohexose-1,6-BP aldolase and 1-phosphorylated fructose kinase, *E. coli* BL21(DE3) cells harboring each plasmid were grown in LB medium containing 50 $\mu$g mL$^{-1}$ Kan at 37°C at an OD$_{600}$ of 0.5 to 0.6. After induction by 1 mM IPTG, the cells were grown for an additional 8 h and harvested by centrifugation at 1,500 × $g$ for 20 min. Expression was analyzed by sodium dodecyl sulfate (SDS)-12% polyacrylamide gel electrophoresis (PAGE) and visualized by staining with Coomassie blue. pET-22b-*araA* was constructed and expressed on the YF1$^{BL\_GatY}$ strain (Fig. S7A and Table S2). The YF1$^{BL\_GatY/EC\_AraA}$ strain was cultured in 0.45% (wt/vol) D-Gal as the primary substrate for AI and 0.05% (wt/vol) D-Fru as a carbon source to support minimal bacterial growth (OD$_{600}$ of 0.5) for IPTG induction during the early stage of growth.

For the purification of recombinant enzymes, centrifuged cells were resuspended in 50 mL of 1× His-binding buffer (20 mM Tris-HCl, 500 mM NaCl, 5 mM imidazole, pH 7.9) and disrupted by sonication. The lysate was centrifuged at 10,000 × $g$ for 20 min to remove cell debris, and the soluble fraction was passed through a 0.45-$\mu$m-pore-size filter. The filtrate was loaded on a His-Bind resin (Novagen) column (10 mL) equilibrated with the same buffer. After washing the column with the washing buffer (20 mM Tris-HCl, 500 mM NaCl, 60 mM imidazole, pH 7.9), 250 mM imidazole was applied to elute the recombinant protein. The fractions containing the enzyme were pooled and dialyzed against 50 mM Tris-HCl buffer (pH 7.0).

**Resequencing.** Genomic DNA (gDNA) of ALE-induced strains was extracted using the Wizard Genomic DNA purification kit (Promega, Madison, WI, USA) and sequenced using the HiSeq 2500 or the NovaSeq 6000 S4 platform (Illumina, San Diego, CA, USA). Whole-genome libraries were generated by TruSeq nano DNA library construction for the Illumina sequencing platform. Paired-end reads were obtained after adapter trimming and quality-filtering using Trimmomatic version 0.39 (50) and mapped to a reference genome (NC_012971.2) using Burrows-Wheeler Aligner (BWA) (51). Single-nucleotide polymorphisms (SNPs) and indels were obtained by variant calling using SAMtools with the parameters minDP5 and minQ 30 (50).

**Genome editing.** Two genome-editing tools were used, the CRISPR-Cas9 system (52) and the $\lambda$ Red recombination method (53). pCas (research resource identifier RRID:Addgene_62225) and pTargetF (RRID: Addgene_62226) plasmids (52) were used to generate SNPs or indels in *E. coli* BL21(DE3), resulting in CraBS_$\Delta$67, FruK_A39S, $\Delta$agaR, and AgaR$^{PTC}$ strains derived from resequencing data. pRedET, FLP plasmids, and the FRT-PGK-gb2-neo-FRT template were used to remove the endogenous genes in *E. coli* YF2 or YF1$^{BL\_GatY}$, resulting in $\Delta$fruA YF2, $\Delta$fruK YF2, $\Delta$kbaY YF2, and $\Delta$araA YF1$^{BL\_GatY}$. Information for plasmids and primers is provided in Table S2.

**Quantitative real-time PCR.** Total RNA was isolated from bacterial samples grown in M9 medium supplemented with each carbon source using an RNeasy minikit (Qiagen, Hilden, Germany). All samples were ground in a mortar under liquid nitrogen before RNA extraction. Genomic DNA was eliminated by RNase-free DNase I treatment during the isolation procedure, and RNA extraction was then performed. RNA quantity and quality were determined by measuring absorbance at 260 nm. All RNA samples with an $A_{260/280}$ ratio of >1.8 were chosen for reverse transcription (RT)-PCR using the iScript cDNA synthesis kit (Bio-Rad, Hercules, CA, USA). The final mixture (20 $\mu$L) was incubated at 25°C for 5 min and 46°C for 20 min, followed by heating at 95°C for 1 min for inactivation. Most cDNA samples were diluted to 10 ng $\mu$L$^{-1}$ prior to PCR. Primers used for qRT-PCR are listed in Table S4. RNA polymerase sigma factor *rpoD* was used as an internal control for normalization. PCR was performed in a CFX Connect real-time system (Bio-Rad) at 95°C for 30 s, followed by 40 cycles at 95°C for 10 s and 55°C for 10 s. Each PCR mixture (10 $\mu$L) contained 10 ng cDNA, 250 nM each primer, and 5 $\mu$L iQTM SYBR green supermix (Bio-Rad).

**Culture conditions and sequencing.** The subcultured cells from 0G to 9G and 17G were spread on M9 medium containing D-Fru (0.05%) and D-Tag (0.45%). A single colony was isolated from the solid medium and inoculated into 96-well plates with M9 liquid medium supplemented with D-Fru (0.05%), D-Tag (0.45%), 0.2 mM IPTG, and 50 $\mu$g mL$^{-1}$ Kan in a total volume of 200 $\mu$L. Each generation of 96-well plates was incubated at 37°C for 12 h, and optical cell density was measured using a BioTek ELx808 absorbance plate reader. Three genetic regions, *fruK*, the Cra binding site, and T7RNAP$^{Pro}$, were amplified from 96-well plates by PCR, and PCR fragments were analyzed by DNA sequencing.

**Enzyme activity assay.** For the aldolase activity assay, D-Fru- 1,6-BP and D-Tag- 1,6-BP (Sigma) were used as substrates. Specific aldolase activities from *E. coli* and *B. licheniformis* were also measured by coupling formation via triosephosphate isomerase (TPI) and glycerophosphate dehydrogenase type I (G3PDH) (54). The assay contained 50 mM Tris-HCl (pH 8.0), 10 mM MgCl$_2$, 0.2 mM NADH, 2 mM ATP, 1 mM substrates, 10 $\mu$g TPI, and 10 $\mu$g G3PDH. Enzymatic activity was monitored spectrophotometrically at 340 nm for 12 min using a microplate reader (Tecan Infinite M200Pro) in a total volume of 200 $\mu$L at 25°C.

For the activity of 1-phosphofructokinase, WT FruK and three variants, D-Fru- 6-P, D-Tag-6-P, (Sigma) and synthesized D-Fru- 1-P (Fig. S8A and B), were used as substrates. Kinase activity was measured by coupling the formation of D-Tag- 1,6-BP or D-Fru- 1,6-BP to NADH oxidation (54) with purified class II fructose 1,6-BP aldolase from *E. coli* (FbaA) for D-Fru-1P and D-Fru-6P or class II D-Tagatose 1,6-BP aldolase from *B. licheniformis* (GatY) for D-Tag-6-P, TPI, and G3PDH. The assay contained 50 mM Tris-HCl (pH 8.0),

10 mM MgCl$_2$, 0.2 mM NADH, 2 mM ATP, 10 $\mu$g TPI, 10 $\mu$g G3PDH, 10 $\mu$g FbaA or GatY, and 2 mM substrates. Enzymatic activity was monitored spectrophotometrically at 340 nm for 12 min, using a microplate reader (Tecan Infinite M200Pro), in a total volume of 200 $\mu$L at 25°C.

AI activity was assayed by measuring the increase in L-ribulose (D-Tag) from L-Ara (D-Gal). Unless otherwise noted, the standard reaction mixtures (1.25 mL) contained 50 mM Tris-HCl buffer (pH 7.0), 0.2 mL of enzyme preparation at a suitable dilution, 1 mM MnCl$_2$, and 0.04 M L-Ara (0.1 M D-Gal). The mixtures were incubated at 40°C for 10 min (60 min), and the reactions stopped by chilling on ice. L-Ribulose (D-Tag) was quantified by the cysteine-sulfuric acid-carbazole method (55). One unit of isomerase activity was defined as the amount of enzyme that produces 1 $\mu$mol product per min under the assay conditions.

To determine the kinetic parameters of EC_AraA and its variants, enzymatic assays were performed as described above, except that AIs were assayed over 1 min to obtain the initial reaction rates. The concentration ranges of L-Ara (D-Gal) were 0 to 40 mM (0 to 400 mM). Kinetic results were obtained by fitting the data to a Michaelis-Menten equation using the software Origin 8.0 (OriginLab Co., Northampton, MA, USA).

Conversion of aldose to ketose by AI was estimated by using high-performance liquid chromatography (HPLC). The reaction mixture (1 mL) contained 50 mM Tris-HCl buffer (pH 7.0), 1 mM MnCl$_2$, 10 mM D-Gal, and the enzyme (1 mg/mL). After incubation at 40°C for 0, 12, 24, 48, 72, or 96 h, the reaction was stopped by cooling on ice. The mixture was centrifuged at 15,000 $\times$ $g$ for 10 min, and an aliquot (20 $\mu$L) was analyzed using a Waters Alliance HPLC system (model number 2695; Waters, Milford, MA, USA) equipped with a refractive index detector (Waters 410) and a CARBOSep CHO-620 column (300 by 6.5 mm, 10-$\mu$m particle size; Transgenomic, Inc., Omaha, NE, USA). The column and refractive index (RI) detector temperatures were 90°C and 35°C, respectively. Deionized water was used as the mobile phase at a flow rate of 0.3 mL/min. The amount of sugar produced was estimated from the ratio of peak areas, based on the calibrated standard curves for D-Gal and D-Tag.

**Synthesis of fructose-1-phosphate.** To a mixture of commercially available dihydroxyacetone phosphate hemimagnesium salt hydrate (50 mg, 0.27 mmol) in Tris-HCl buffer (50 mM, 2 mL), D-glyceraldehyde (74 mg, 0.83 mmol) was added, followed by 200 U of rabbit muscle aldolase (2 mg). The resulting suspension was stirred at 30°C (48 h). Ethyl acetate (EtOAc; 5 mL) and H$_2$O (5 mL) were added to the reaction mixture. The combined aqueous layers were concentrated, and methanol (MeOH) was added. When the solid formed, it was filtered with cold MeOH to obtain the desired compound (5 mg, 7%, ivory solid). Retention factor (Rf) = 0.11 (EtOAc/MeOH/distilled water/acetic acid 5/2/1.4/0.4); $^1$H nuclear magnetic resonance (NMR) (D$_2$O, 400 MHz) $\delta$ 3.59–4.17 (m, 7H); 13C NMR (D$_2$O, 100 MHz) $\delta$ 97.91 (d, $J$ = 7.8 Hz), 69.37, 69.11, 67.72, 66.36, 63.39.

**3-D structural modeling.** Highly accurate 3-D protein structures of EC_FruK WT (WP_000091263.1) and three variants, the A39S, V232F, and E236G mutants, were predicted using AlphaFold version 2.2.1 (56). For cavity detection-guided blind docking, the molecular ring structures of phosphorylated sugars derived from PubChem database were used. Molecular docking was carried out using the CB-Dock online platform (57). The top-ranked docking model was selected for visual analysis by using PyMOL.

**Generation of an *E. coli araA* random mutagenesis library.** Genomic DNA was isolated from *E. coli* MG1655 using a Genomic DNA extraction kit (Qiagen) according to the manufacturer's instructions. The *araA* gene was amplified by PCR from *E. coli* genomic DNA using EC_AraA NdeI_F and 10 pmol of primer HindIII_R primers. The PCR mixture (50 $\mu$L) contained 20 ng of genomic DNA, 10 pmol each primer, 1 $\times$ PCR buffer, 0.2 mM dNTP mix, and 2.5 U of PrimeSTAR HT DNA polymerase. The PCR product was cloned into the pTOP Blunt V2 vector, and the resulting construct was transformed into *E. coli* DH5$\alpha$ competent cells. Transformants containing pTOP Blunt V2 harboring the *araA* gene were selected on LB-Amp plates. Plasmid DNA isolated from the transformants was digested with NdeI and HindIII and ligated into the NdeI and HindIII sites of pET-22b(+) (Novagen), yielding pET-22b(+)-EC_AraA. Expression of the recombinant enzyme and validation were performed following the methods for D-tagatose-1,6-BP aldolase described above, using Amp instead of Kan. For the construction of a random mutagenesis library, mutations were introduced into the *araA* gene by error-prone PCR, using a Clontech Diversify PCR random mutagenesis kit (Mountain View, CA, USA). The mutagenesis experiment generated an error frequency of ~5 base substitutions per 1,000 bases per gene copy, as described previously (58). The libraries created were ligated into pET-22b(+) between NdeI and HindIII restriction sites to yield pET-22b(+)-EC_AraA ML.

**Screening of EC_AraA variants from the library using the D-tagatose auxotrophic *E. coli* system.** To screen EC_AraA variants that can complement the D-tagatose auxotrophic strain, recombinant *E. coli* BL21(DE3) $\Delta araA$ harboring pET-28a(+)-BL_TBA competent cells were transformed with *E. coli araA* genetic libraries [i.e., pET-22b(+)-EC_AraA ML] by electroporation. *E. coli* BL21(DE3) $\Delta araA$ transformants harboring pET-28a-BL_TBA and pET-22b(+)-EC_AraA ML were grown for 8 to 12 h on LB plates containing 50 $\mu$g of Amp and 25 $\mu$g of Kan per mL. To collect bacterial cells, LB medium (1 mL) was poured onto the agar plates, and the resuspended colonies were cultured at 37°C for 1 h. The culture was washed twice with M9 medium. The washed cells (OD$_{600}$ of 0.1) were inoculated into freshly prepared M9 medium (50 mL) containing appropriate antibiotics, 0.2 mM IPTG, 0.05% D-Fru, and 0.45% D-Gal. When cells had grown to an OD$_{600}$ of 1.8 to 2.0, the culture was transferred to the fresh medium for several rounds of cultivation until cellular growth reached the growth rate of the original culture. Each transferred culture was poured onto LB-Amp-Kan agar plates containing 50 $\mu$g of Amp and 25 $\mu$g of Kan per mL. After each pop-up colony was grown in 3 mL of LB medium, pET-22b(+)-EC_AraA ML plasmids were extracted and verified by DNA sequencing.

**Validation of EC_AraA variants.** Several EC_AraA mutants with increased substrate specificity to D-Gal were selectively isolated, and each gene encoding an EC_AraA mutant was cloned into the pET-15b(+) plasmid and expressed in *E. coli* BL21(DE3). Briefly, PCR was performed to amplify the *araA* mutant gene from

the library using two primers, EC_AraA NdeI_F2 and EC_AraA XhoI_R. The PCR product was cloned into the pTOP Blunt V2 vector and transformed into *E. coli* DH5α competent cells. Transformants containing the pTOP Blunt V2 vectors harboring the *araA* mutant genes were selected on LB-Amp plates. Plasmids were isolated from the transformants and digested with NdeI and XhoI. Inserts were ligated into the NdeI and XhoI sites of the pET-15b(+) plasmid (Novagen), yielding pET-15b-EC_AraA, pET-15b-EC_AraA_H17R/R159S/V168A, pET-15b-EC_AraA_E22D/M95L/H157L, and pET-15b-EC_AraA_V368A/E493D, respectively, which were transformed into competent *E. coli* BL21(DE3) Δ*araA* cells. The expression vectors also encoded an N-terminal polyhistidine (6 × His) tag in-frame with the inserted gene. Expression of the wild-type EC_AraA and its variants was performed as described above.

For the purification of recombinant enzymes, as described elsewhere (59, 60), centrifuged cells were resuspended in 50 mL of 1 × His-binding buffer and disrupted by sonication. The lysate was centrifuged at 10,000 × *g* for 20 min to remove cell debris, and the soluble fraction was passed through a 0.45-μm-pore-size filter. The filtrate was loaded on a His-Bind resin (Novagen) column (10 mL) equilibrated with the same buffer. After washing the column with the washing buffer (20 mM Tris-HCl, 500 mM NaCl, 60 mM imidazole, pH 7.9), 250 mM imidazole was applied to elute the recombinant protein. The fractions containing enzyme were pooled and dialyzed against 50 mM Tris-HCl buffer (pH 7.0). Protein concentrations were determined by the bicinchoninic acid method (61). Enzyme fractions were analyzed by 12% SDS-PAGE and visualized with Coomassie blue. To verify the conformational integrity of EC_AraA variants, circular dichroism (CD) was determined using a Jasco J-810 spectropolarimeter with a Peltier temperature-controlled cuvette holder as described previously (59).

**Data availability.** Sequence Read Archive (SRA) data generated in this study have been deposited in the NCBI database under BioProject accession number PRJNA897095 for *E. coli* BL21(DE3).

## SUPPLEMENTAL MATERIAL

Supplemental material is available online only.
**SUPPLEMENTAL FILE 1**, PDF file, 5.1 MB.

## ACKNOWLEDGMENTS

This work was partly supported by the National Research Foundation (NRF) of Korea (grant number 2021M3A9I4021431), funded by the Ministry of Science, ICT, and Future Planning (MSIP), by the Technology Innovation Program (grant number 20015807) funded by the Ministry of Trade, Industry & Energy (MOTIE, Korea), and by a grant (grant number HP20C0082) from the Korea Health Technology R&D Project through the Korea Health Industry Development Institute (KHIDI), funded by the Ministry of Health & Welfare, Republic of Korea.

Y. Joo, J.-Y. Sung, S.-M. Shin, and D.-W. Lee formulated the research plan. Y. Joo, J.-Y. Sung, S.-M. Shin, S. J. Park, K. S. Kim, and K. D. Park performed the experiments. Y. Joo, J.-Y. Sung, S.-M. Shin, K. D. Park, S.-B. Kim, and D.-W. Lee analyzed the data. Y. Joo, J.-Y. Sung, and D.-W. Lee wrote the manuscript. D.-W. Lee conceived, planned, supervised, and managed the study.

We declare no conflict of interest.

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
