## [Reviewer comments · Microbiology Spectrum]

Microbiology Spectrum

A retro-aldol reaction prompted the evolvability of a phosphotransferase system for the utilization of a rare sugar

Yunhye Joo, Jae Yoon Sung, Sun-Mi Shin, Sun Jun Park, Kyoung Su Kim, Ki Duk Park, Seong-Bo Kim, and Dong-Woo Lee

Corresponding Author(s): Dong-Woo Lee, Yonsei University

Review Timeline:

Submission Date:	September 9, 2022
Editorial Decision:	October 7, 2022
Revision Received:	November 5, 2022
Editorial Decision:	November 20, 2022
Revision Received:	December 15, 2022
Editorial Decision:	January 2, 2023
Revision Received:	January 15, 2023
Accepted:	January 25, 2023

Editor: Jing Han

Reviewer(s): The reviewers have opted to remain anonymous.

Transaction Report:

DOI: <https://doi.org/10.1128/spectrum.03660-22>

October 7, 2022

Prof. Dong-Woo Lee
Yonsei University
Biotechnology
Yonseiro 50
Seoul 03722
Korea (South), Republic of

Re: Spectrum03660-22 (A retro-aldol reaction prompted the evolvability of the phosphotransferase system to utilize rare sugars)

Dear Prof. Dong-Woo Lee:

"Data Availability Statement" is missing in the "Materials and Methods" section.

Link Not Available

Sincerely,

Jing Han

Journals Department
Reviewer comments:

Reviewer #1 (Comments for the Author):

This is very nice work. The authors successfully obtained E. coli mutants to utilize the rare sugar, and then took a systematic approach to determine the evolutionary trajectory of E. coli mutants to utilize the rare sugar. The mechanism by which E. coli evolved rare sugar utilization can initiate the utilization of new sugars. In my opinion, it adds valuable data to our understanding of sugar utilization in E. coli.

Reviewer #2 (Comments for the Author):

The authors have produced an *E. coli* B (BL21(DE3)) strain capable of growing on the rare ketose sugar tagatose. They start with an *E. coli* B strain BL21(DE3) in which they express aldolases capable of cleaving tagatose-1,6-phosphate as a crucial step in tagatose metabolism. They isolate a 1st strain which grows well on tagatose in the presence of the plasmid and from this, another strain no longer requiring the plasmid. Sequencing both strains identified mutations causing the overproduction of the fruBAK operon and a point mutation in fruK allowing further metabolism of tagatose to tagatose 1,6-P, the substrate of the aldolase. In the second strain, without the plasmid, an agaR mutation derepressed the expression of kbaYZ genes encoding an authentic *E. coli* tagatose 1,6P aldolase. They have analysed three fruK mutations which appeared intermediately but only one of which fixed, and found lower activity towards the authentic substrate and some activity on alternative substrates. Subsequently they used their tagatose-utilising strain to select mutations in arabinose isomerase which allow better activity on galactose converting it to tagatose. They propose their system as a platform for engineering sugar isomerases.

E. coli strains cannot grow on tagatose but some *Klebsiella* and *Salmonella* can. Shakeri-Garakani et al 2004 had already produced an *E. coli* capable of use of tagatose by adding a plasmid expressing the *Klebsiella* tag genes in *E. coli*

They are not the only people interested in this sugar a recent paper Al'Abri et al NAR 2022 have also used directed evolution to isolate *E. coli* capable of growing on tagatose. This paper should be cited and methodologies compared.

The paper is not always easy to follow, lots of things need to be explained to make it easily comprehensible to the average reader. In particular for the selection procedure.

Retro - aldol reaction needs to be explained: it is a chemical term, reverse of a synthetic aldol condensation. Biochemists and molecular biologists are more familiar with the enzyme aldolase which performs the reaction and they should explain this. In fact they start with genes expressing known aldolases so the retro-aldol term just adds an unnecessary complication.

L. 112, 339 etc "Rare sugar-rich conditions", "Rare sugar-utilizing strains" etc are ambiguous. The authors want to say (rare sugar)-rich conditions and (rare sugar)-utilizing strains but in fact it reads as rare "sugar-utilizing" strains i.e. as if strains utilizing sugars are rare! To be clear the authors need to write "strains utilizing rare sugars" etc

L. 85 rare sugar D-Tag, including D-Gal did not support *E. coli* BL21(DE3) growth? This phrase is not clear.

Also L. 274

The authors need to explain clearly that the *E. coli* strain they are using (BL21(DE3)) is an *E. coli* B and, unlike many other *E. coli* strains (e.g. the standard laboratory K12), does not grow on galactose (apparently due to the loss of some genes in the galactose degradation pathway (known as the Leloir pathway (L.274).

Did they try their protocol in a Gal+ strain or is the delta gal implicated?

The logic of their plan in this first paragraph is not clear. They start by saying that probably BL21 is missing the PTS genes for the transport of tagatose that are found in other Tag+ bacteria. Then they switch to talk about aldolases! They need to explain that in order to select for mutations leading to evolution of existing PTS genes to transport an alternative substrate they decided to overexpress in the bacteria a downstream gene required for tag metabolism. They should thus explain the expected degradation pathway and the choice of enzymes. Fig. 1 shows the gat (galacticol) PTS operon but its role and the presence of endogenous tagatose-1,6-P aldolase is not mentioned in the text although they do test the purified enzymes in vitro.

L.94,95 This sentence is only clear to people who know that the KbaY and GatY aldolases are heterodimeric and only fully active with the subunits KbaZ and GatZ. At least they should add "required for full activity" to their sentence before reference (24). Why did not they include both subunits in their overproduction purification test?

BL21(DE3) is used to express proteins for overproduction and purification from a plasmid (e.g. pET28a in their case) under control of the T7 promoter and usually produces large amounts of the protein in the cell. They are using it for their ALE with a high amount of IPTG presumably completely changing cellular physiology. Did they look at the protein profile of their strains? Did they think to try with a more physiological level of the cloned aldolase? The mutation in the lacUV5 promoter was crucial to reduce expression of the BL_GatY and hence might have reduced the time required to obtain YF1, if they had used a lower expression system. To this reader it does not seem to be the best strain to start with. Maybe they wanted the Gal minus to isolate arabinose isomerase mutants?

Have others isolated AraA mutants with improved catalysis of galactose? How do their araA mutations compare to these?

L. 101, 409 they use 0.05% fructose as "minimum (should be minimal) helper substrate" which will induce the fru genes including fruA the fructose PTS transporter which might be expected to transport another ketose sugar like tagatose or evolve to a new or wider specificity. They actually say that FruBA might function as a D-tagatose transporter (L.112-113) but it seems out of place here as they next talk of curing the BY_GatY plasmid expressing the aldolase, without explaining why. As the cured strain (YF1) did grow slowly on 0.05% fru+0.45% tagatose they conclude it had acquired mutations allowing utilization of tagatose, which could be in the expressed fructose PTS. Importantly it only took one additional transfer (if I've understood Fig. 2A correctly) to get YF2 which grew as well as YF1 with the BY_GatY plasmid. Interestingly they did not isolate mutations in the Fructose PTS genes themselves. This could either be because it is equally efficient at transporting fructose and tagatose or the

high expression of the operon from the loss of Cra/FruR repression compensated for a lower affinity for tagatose. Was the presence of the 0.05% fructose with 0.45% tagatose important for their selection? Did they compare with glucose or another sugar? It seems that the way that their selection was set up with low fructose in the media they were expecting/directing that the pathway would be via the fructose operon.

L.124, 132 On reading it I wanted to know here what is the predicted effect of the lacUV5 promoter mutation expressing the T7 polymerase and hence its possible effect on BY_GatY expression from the T7 promoter?
L. 180 maybe gives the answer, the mutation changes the lacUV -10 promoter to the original lac promoter TATAAT to TATGTT. However this does not agree with the GA to AT change listed as the lacUV5 mutation in Table S1). In any case they should say it is a mutation reducing the T7 expression here and refer to further discussion later, if necessary.

Compared to the limited details of the selection the analysis of the mutations obtained is rather long and also not always easy to follow (L.138 -222). This analysis would be more appropriate if they had several different mutations to compare rather than the three mutations expected from such a selection.

Other points:

L.30 Peripheral PTS ? explain

L. 91, L. 428 Class II aldolase Is this important? What is difference from class I ?

L. 99 Not "Subsequently" better to say "In order to achieve this"

L.106 Replace "in over 500h" with "after 500h"

L.108 Serial transfer culture? Presumably cultureS. Explain of from what to what

L.123 Cra is also known as FruR and it is its role as the fructose-specific repressor which is important here. The authors should state that Cra is FruR and the delta67 is removing the Cra/FruR binding site and derepressing the fruBAK operon.

L.142 onwards please refer to the figures where these results are shown. The text is not easy to follow. None of their growth curves give the original BL21(DE3) for comparison.

L.155 a phenotype e.g. growth on fructose is not REversed by growth on another sugar (which is a different phenotype).

L.158 loss of a specific regulator (AgaR) removes a sugar-operon specific repression and not general catabolite repression (generally due to cAMP, CAP and EIIA of the PTS)

L.160 Their nomenclature of the strains carrying reconstructed mutations in BL21 is not orthodox but I appreciate their attempt to produce simple names for multimutated strains. The use of capital letters for the genes mutated and for the number of genes mutated is confusing and it might be easier if they use lower case letters for the in the single, double, triple (e.g. triple mutation tCFA). In theory the s, d and t are redundant with the number of letters in the name (e.g CFA is a triple mutation).

L.163 the growth curves for TCFA and TCFA(PTC) Fig. S2F are very different. Any comment?
Surprisingly some of the combinations of mutations do affect growth on glucose.

L.181reduced to that of the original.... The sentence is badly constructed. The mutation in the T7 RNAP promoter changed the -10 of the promoter from (... TATAAT) of the high expression lacUV5 promoter to .. (TATGTT) corresponding to the original lac promoter and should thus have reduced expression of the T7 polymerase and of its downstream targets. (Was this verified?)

L. 186 Fig.S4A does not seem to show lag times. Please verify that the labels on the abscissa are correct.
What does "except for D-Fru in a plasmid dependent manner" refer to ? Do they mean "did not exhibit a lag time on glucose or fructose, except for Y1 in the absence of the GatY plasmid"??

L.198 a verb is missing - "was >2-fold higher"

L.211 Fig.S5C Why should deletion of fruK (specific for fructose-1P) reduce growth on glucose in YF2?

L.213 "located in GalN PTS and galactitol PTS" Do they mean that kbaY is required for the metabolism of sugars transported by two different PTS. The kbaY gene is located with genes for the galactitol PTS (Fig.1A).

L. 220 It is only novel in E. coli but a similar pathway exists already in Salmonella and Klebsiella!

L.230 Fig. 3D What is the bar diagram to the right?

Fig.3F What is the diagram to the right?

L.260 fits well

L.271 we need some preliminary information. Please say that arabinose isomerase can catalyse the formation of tagatose from galactose (ref). They take advantage of the fact that BL21(DE3) is missing genes of the Leloir pathway to show cloned arabinose isomerase can allow growth on galactose via the tagatose utilization pathway.

L.282 what do they mean "grow solely by enzymatic activity"?

L. 291 YFI (BL_GAT/EC_Ara ML) Their strain contain two pET plasmids with different antibiotic resistances but are they the same origin of replication? In which case the copy numbers of the two high copy plasmids are going to vary. 50 microg/ml of ampicillin will soon be used up in any case.

Fig. 4D and 4F why is the lag for use of galactose so much longer in 4F than 4D? Fig. 4D should have the AraA plasmid indicated like the araA mutation in 4C.

L. 414 "without antibiotics, and an inducer" reads as if it is without antibiotics but with an inducer. I suppose they mean "without antibiotics and IPTG" i.e. without the inducer

L.555 "mutation(s) were introduced" it is unlikely they get a single mutation

L.570 when cells HAD grown

L.571 I do not understand why "growth was not faster than the growth rate of the original culture"?

L.734 what is the "clinker strategy"?

L.752 filled grey circle

L.764 Analysis of the effects of

L.783 Fig. 4B Why is the abscissa labelled as time on a histogram showing specific activity?

Staff Comments:

Preparing Revision Guidelines

Please return the manuscript within 60 days; if you cannot complete the modification within this time period, please contact me. If you do not wish to modify the manuscript and prefer to submit it to another journal, please notify me of your decision immediately so that the manuscript may be formally withdrawn from consideration by Microbiology Spectrum.

Response to Reviewers (Spectrum03660-22)

□ Editor

"Data Availability Statement" is missing in the "Materials and Methods" section.

► **We included it in the revised text (L616).**

Please clearly label your introduction "Introduction."

► **We labeled "Introduction" (L51).**

Please callout supplementary table s4 in the manuscript.

► **We revised the text for "Table S4" (L216 and L490) and "Table S3" (L437).**

□ Reviewer #1 (Comments for the Author):

This is very nice work. The authors successfully obtained E. coli mutants to utilize the rare sugar, and then took a systematic approach to determine the evolutionary trajectory of E. coli mutants to utilize the rare sugar. The mechanism by which E. coli evolved rare sugar utilization can initiate the utilization of new sugars. In my opinion, it adds valuable data to our understanding of sugar utilization in E. coli.

► **We thank you so much for your nice appreciation for our paper.**

□ Reviewer #2 (Comments for the Author):

The authors have produced an E. coli B (BL21(DE3)) strain capable of growing on the rare ketose sugar tagatose. They start with an E. coli B strain BL21(DE3) in which they express aldolases capable of cleaving tagatose-1,6-phosphate as a crucial step in tagatose metabolism. They isolate a 1st strain which grows well on tagatose in the presence of the plasmid and from this, another strain no longer requiring the plasmid. Sequencing both strains identified mutations causing the overproduction of the fruBAK operon and a point mutation in fruK allowing further metabolism of tagatose to tagatose 1,6-P, the substrate of the aldolase. In the second strain, without the plasmid, an agaR mutation derepressed the expression of kbaYZ genes encoding an authentic E. coli tagatose1,6P aldolase. They have analysed three fruK mutations which appeared intermediately but only one of which fixed, and found lower activity towards the authentic substrate and some activity on alternative substrates. Subsequently they used their tagatose-utilising strain to select mutations in arabinose isomerase which allow better activity on galactose converting it to tagatose. They propose their system as a platform for engineering sugar isomerases.

► **We greatly appreciate your critical specific comments, which help us to improve our manuscript clearer to the potential readers. Please, see our point-by-point responses to your specific comments below.**

E. coli strains cannot grow on tagatose but some Klebsiella and Salmonella can. Shakeri-Garakani et al. 2004 had already produced an E. coli capable of use of tagatose by adding a plasmid expressing the Klebsiella tag genes in E. coli. They are not the only people interested in this sugar a recent paper Al'Abri et al. NAR 2022 have also used directed evolution to isolate E. coli capable of growing on tagatose. This paper should be cited and methodologies compared.

► **As advised, we cited those papers as Refs. #22 (Shakeri-Garakani et al. 2004) and #24 (Al'Abri et al. NAR 2022), respectively, in the revised text (L88, L90). As you mentioned, both papers described the generation of E. coli capable of D-tagatose (D-Tag) through the heterologous expression of tag-specific PTS genes from Klebsiella and Bacillus licheniformis. In fact, Shakeri et al. cloned a series of D-Tag PTS genes corresponding to gatY-tagKTH of Klebsiella oxytoca**

into a plasmid pLAS1 to confer *E. coli* strain to use D-Tag, indicating that the gene cluster is responsible for D-Tag utilization through combinatorial deletion analysis of plasmid. Also, “Al'Abri et al. (2022)” provided additional information on tag-specific PTS. The authors integrated the D-Tag operon from *B. licheniformis* ATCC 14580 into *E. coli* using phagemid, and performed adaptive evolution to enhance the bacterial use of D-Tag. In fact, both groups used the entire D-Tag PTS genes from Tag+ phenotype strains and implemented those in *E. coli* as a model strain to identify their function and induced further directed evolution to obtain *E. coli* mutants with D-Tag+ phenotype.

In the present study, however, we introduced a foreign *gatY* gene from *B. licheniformis* to *E. coli*, resulting in the occurrence of adaptive evolution to yield D-Tag+ *E. coli* mutant strains, and investigated how to gain the D-Tag preference through endogenous genes in the core metabolic pathways. Such an approach enabled us to reveal a new evolutionary path to gain promiscuous sugar preference via adaptive evolution primarily occurring in the fructose-specific *fruBKA* operon, which was not observed in both studies. This result reveals that the Fru PTS may evolve to use two different ketoses (i.e., D-Tag as well as D-Fru), implying the resilience of enteric bacteria in preferable sugar-limited environments. Furthermore, we found that tagatose usability was enhanced through additional mutations in endogenous regulators and non-coding regions for fine-tuned expression levels, even in the absence of a foreign aldolase gene. This intriguing result demonstrates that the evolved strain by *fruBKA*-based adaptation utilizes D-Tag. To clarify this issue, we rephrased the relevant parts in the revised text (L88-90)

The paper is not always easy to follow, lots of things need to be explained to make it easily comprehensible to the average reader. In particular for the selection procedure.

► As advised, we rephrased and explained the relevant parts throughout the paper to make them easily comprehensible to the average reader (See the specific responses to your comments below).

Retro-aldol reaction needs to be explained: it is a chemical term, reverse of a synthetic aldol condensation. Biochemists and molecular biologists are more familiar with the enzyme aldolase which performs the reaction and they should explain this. In fact, they start with genes expressing known aldolases so the retro-aldol term just adds an unnecessary complication.

► As advised, we explained the retro-aldol reaction to retro-aldol cleavage reactions derived from sugar aldolase (L110-111).

L. 112, 339 etc "Rare sugar-rich conditions", "Rare sugar-utilizing strains" etc are ambiguous. The authors want to say (rare sugar)-rich conditions and (rare sugar)-utilizing strains but in fact it reads as rare "sugar-utilizing" strains i.e., as if strains utilizing sugars are rare! To be clear the authors need to write "strains utilizing rare sugars" etc

► As advised, we revised these terms throughout the text (L77, L79, L125, L356, and L381).

L. 85 rare sugar D-Tag, including D-Gal did not support E coli BL21(DE3) growth? This phrase is not clear.

► To clarify this, we rephrased this sentence (L86).

Also L. 274 The authors need to explain clearly that the E. coli strain they are using (BL21(DE3)) is an

E. coli B and, unlike many other *E. coli* strains (e.g. the standard laboratory K12), does not grow on galactose (apparently due to the loss of some genes in the galactose degradation pathway (known as the Leloir pathway (L.274).

► As you advised, we clearly explained the genotype and phenotype of *E. coli* strain used in this study (L291).

Did they try their protocol in a Gal⁺ strain or is the delta gal implicated?

► We tried to use a Gal⁻ *E. coli* BL21(DE3) strain lacking *galTKE*⁻ because we sought to obtain a platform cell for screening tailor-made L-arabinose isomerase (AI) for the production of D-tagatose. For this reason, we required the strain with Gal negative phenotype (optionally including the *ompT* and *lon*⁻) B strain with T7 promoter, which facilitates us to express recombinant AI mutant library plasmid.

The logic of their plan in this first paragraph is not clear. They start by saying that probably BL21 is missing the PTS genes for the transport of tagatose that are found in other Tag⁺ bacteria. Then they switch to talk about aldolases! They need to explain that in order to select for mutations leading to evolution of existing PTS genes to transport an alternative substrate they decided to overexpress in the bacteria a downstream gene required for tag metabolism. They should thus explain the expected degradation pathway and the choice of enzymes. Fig. 1 shows the gat (galactitol) PTS operon but its role and the presence of endogenous tagatose-1,6-P aldolase is not mentioned in the text although they do test the purified enzymes in vitro.

► To clarify the logic of our plan in the first paragraph you raised the issues, we have rephrased the text (L88-L98).

L.94,95 This sentence is only clear to people who know that the KbaY and GatY aldolases are heterodimeric and only fully active with the subunits KbaZ and GatZ. At least they should add "required for full activity" to their sentence before reference (24).

► As you advised, we revised the relevant text (L 106).

Why did not they include both subunits in their overproduction purification test?

► It is noted that all class II tagatose-1,6BP aldolases except *B. licheniformis* (GatY; ~30 kDa) should have heterodimeric complexes (GatY; ~30 kDa, GatZ; ~47 kDa) for full activity (Andreas Brinkkötter et al. Arch. Microbiol. 2002) (L106). Nevertheless, we did not include both subunits because we concerned about the metabolic burden and the occurrence of protein aggregation and/or misfolded protein by expressing both genes during a long evolution period. In addition, although characterization of GatYZ and KbaYZ has been performed in vitro (Andreas Brinkkötter et al. Arch. Microbiol. 2002), to our best knowledge, there are still no reports of whether the activity and expression levels in cells are sufficient to support cell growth. In fact, WT BL21(DE3) in D-Tag containing M9 medium could not grow (Fig. 1G). Therefore, to accelerate the adaptive evolution, we tried to heterologously express the single subunit of BL_GatY, which has better activity than EC_GatY and EC_KbaY (Fig. 1D), known as tagatose aldolase. Furthermore, the purpose of the aldolase activity assay in the present study was not to identify whether there was an aldolase activity as a complete complex, but to compare and select an appropriate aldolase with the highest aldol cleavage activity as a single subunit. Therefore, we tested EC_KbaY and EC_GatY subunits separately only for their monomeric aldolase activity.

BL21(DE3) is used to express proteins for overproduction and purification from a plasmid (e.g. *pET28a* in their case) under control of the T7 promoter and usually produces large amounts of the protein in the cell. They are using it for their ALE with a high amount of IPTG presumably completely changing cellular physiology. Did they look at the protein profile of their strains? Did they think to try with a more physiological level of the cloned aldolase? The mutation in the *lacUV5* promoter was crucial to reduce expression of the *BL_GatY* and hence might have reduced the time required to obtain YF1, if they had used a lower expression system. To this reader it does not seem to be the best strain to start with. Maybe they wanted the *Gal minus* to isolate arabinose isomerase mutants?

► Yes, we also found that when we used *E. coli* K-12 BW25113 WT strain harboring pBAD-BL_GatY plasmid with a relatively lower expressed promoter than a strong T7lac promoter in *E. coli* BL21(DE3) strain harboring the pET-BL_GatY plasmid (see below unpublished data figure C), we obtained ALE-induced *E. coli* K-12 BW25113 mutant tag⁺ strains after ~200 h (mutant strains were discussed in your answer about basal level media (0.05% D-Fru), and we will deal with the result at a follow-up study). On the other hand, as we answered your comments above regarding the selection of Gal- *E. coli* BL21 (DE3) strain, we aimed to design the ALE-induced strain derived from a Gal- where we introduced the AraA mutant libraries for screening tailor-made enzymes for the conversion of D-Gal to D-Tag. Nevertheless, we also investigated the effect of the pET-BL_GatY plasmid expressed in *E. coli* BL21(DE3) using SDS-PAGE (see below unpublished data figure E), indicating that there was little variations in the whole cell protein patterns of the strain after 500 h evolution.

(Unpublished data)

(Unpublished data) Comparison of the growth profiles for *E. coli* K-12 BW25113 strain harboring pBAD-BL_GatY plasmid (A) vs. *E. coli* BL21 (DE3) strain harboring the pET-BL_GatY plasmid (B) in M9 containing D-Fru as the sole carbon source. (C & D) The effect of single gene deletion in various transporters and PTS related genes on the growth of the *gatY* expressing *E. coli* BW15113 and BL21(DE3) in M9 containing D-Tag as the sole carbon source. (E) SDS-PAGE analysis of the whole cell protein patterns in *E. coli* BL21(DE3) strains.

Have others isolated AraA mutants with improved catalysis of galactose? How do their araA mutations compare to these?

► There are several attempts to improve the catalysis of the industrially promising AraA including our group (Ravikumar et al. Trends Food Sci Technol. 2021). For example, H18T from *Geobacillus stearothermophilus* (Laksmi et al. Biochim. Biophys. Acta-Proteins Proteom. 2018), F279I from *B. coagulans* (Zhaojuan Zheng et al. J. Agric. Food Chem. 2017), F279N from *Shewanella* sp. (Jayaraman et al. Enz. Microb. Technol. 2021). Among them, H18T coincides with H17R from our results, which contributes to better accessibility toward D-Gal bounded to the binding pocket through increased structural flexibility.

L. 101, 409 they use 0.05% fructose as "minimum (should be minimal) helper substrate" which will induce the fru genes including fruA the fructose PTS transporter which might be expected to transport another ketose sugar like tagatose or evolve to a new or wider specificity. They actually say that FruBA might function as a D-tagatose transporter (L.112-113) but it seems out of place here as they next talk of curing the BY_GatY plasmid expressing the aldolase, without explaining why. As the cured strain (YF1) did grow slowly on 0.05% fru+0.45% tagatose they conclude it had acquired mutations allowing utilization of tagatose, which could be in the expressed fructose PTS. Importantly it only took one additional transfer (if I've understood Fig. 2A correctly) to get YF2 which grew as well as YF1 with the BY_GatY plasmid. Interestingly they did not isolate mutations in the Fructose PTS genes themselves. This could either be because it is equally efficient at transporting fructose and tagatose or the high expression of the operon from the loss of Cra/FruR repression compensated for a lower affinity for tagatose.

► As you pointed out, we corrected it (L113) and rephrased the relevant part to explain the purpose of plasmid curing (L128-L129). For your information, we clearly stated that we also re-sequenced YF2 strain the following BL_GatY plasmid curing (L133-L144). As a result, we found that only *agaR* deletion was the main genetic mutation in YF2 strain, which is also mainly described in the results of this study (Fig. 2B and Table. S1).

Was the presence of the 0.05% fructose with 0.45% tagatose important for their selection? Did they compare with glucose or another sugar? It seems that the way that their selection was set up with low fructose in the media they were expecting/directing that the pathway would be via the fructose operon.

► Yes, when we designed the experiment for ALE, we also tested another type of combination (0.05% glucose (D-Glc) and 0.45% D-Tag). We found that either competitive transporter collisions between glucose and tagatose or glucose-induced catabolite repression resulted in the diauxic curve with extended lag time (Data not shown). This is probably due to the structural difference between aldose (D-Glc) and ketose (D-Fru). However, the 4-epimer relationship between D-Fru and D-Tag did not show any problematic growth. Moreover, we investigated the effects of single gene deletion in sugar PTS genes (e.g., D-Fru, D-Glc, D-Man) on bacterial growths of *E. coli* B and K-12 strains harboring BL_GatY plasmid (see the unpublished data above figure A to D). These results demonstrated that only mutant strains with an individual deletion in *fruBKA* did not grow on D-Tag. Accordingly, we induced the expression of the *fru* operon by supplying D-Fru to identify the evolutionary pathway in *E. coli* grown on 0.05% fructose as a minimal helper substrate.

L.124, 132 On reading it I wanted to know here what is the predicted effect of the lacUV5 promoter mutation expressing the T7 polymerase and hence its possible effect on BY_GatY expression from the T7 promoter? L.180 maybe gives the answer, the mutation changes the lacUV -10 promoter to the original

lac promoter TATAAT to TATGTT. However this does not agree with the GA to AT change listed as the *lacUV5* mutation in Table S1). In any case they should say it is a mutation reducing the T7 expression here and refer to further discussion later, if necessary.

- ▶ We corrected mistyped *lacUV5* SNVs in (Table S1). As you described, AA::GT at 749,894..749,895 and G::A at 749,902 (NC_012971.2) mutations occurred in YF strains. These mutations are identical to (ref. 33-35) we cited. As you advised, we mentioned the mutational effect to reduce the T7 expression (L147-L148).

Compared to the limited details of the selection the analysis of the mutations obtained is rather long and also not always easy to follow (L.138 -222). This analysis would be more appropriate if they had several different mutations to compare rather than the three mutations expected from such a selection.

- ▶ As shown in Fig 2, there were six mutations found in the ALE-induced strains, and only four of them, CraBS Δ 67, FruK_A39S, AgaR^{PTC}, and T7RNAP^{pro}, were directly related to the sugar uptake system. We found these mutations are the essential factors for the D-tag availability (L. 133-143). The result part you pointed out deals with how the loss of tight regulation occurred in a sugar-specific PTS to gain sugar usability. First, we verified each of the mutation effects in the growth profile on D-Tag by introduction of single, double, and triple mutations (L159-182). In turn, we tried to identify minimal genetic elements for rare sugar utilization. We analyzed the correlations between genotypic and phenotypic changes by 14 mutant strains with all types of mutant combinations (L183-L210). And we finally figured out how the novel flux was induced and controlled by the minimal mutations (L. 211-239). If we have a chance, we would like to analyze other mutations that are not essential but for enhancing flux stabilization, which we did not refer to in this study.

Other points:

L.30 Peripheral PTS? explain

- ▶ “peripheral PTS, consist of IIA and IIB membrane proteins or domains; sugar-specific energy coupling proteins” is cited from J Mol Microbiol Biotechnol. (2015) 25(0): 73-78. Acceptance of various substrates, which is called the moonlight effect or promiscuity of different types of PTS has been reported a lot (Kornberg HL et al. J Mol Microbiol Biotechnol. 2001; Kornberg et al. Proc Natl Acad Sci U S A. 2000). This means that transporter does not accept only one substrate because sugars have similar structures, suggesting that they can complement each other and fit to the more specific sugar-rich environment through some SNV mutations. These probably evolved by diversifying the possibilities of adaptation rather than being too specialized to a single sugar structure for survival.

L. 91, L. 428 Class II aldolase Is this important? What is difference from class I?

- ▶ Aldolases are classified into three groups: Class-I, Class-IA, and Class-II; all classes share similar structural features but have low amino acid identity. Class-I aldolases utilize a lysine residue within the active site to create a stabilizing Schiff base intermediate with the substrate. Class-I aldolases typically form homotetramers with a total molecular weight of ~160 kDa (Gamblin et al., 1990; Cooper et al., 1996). In contrast, Class-II aldolases are metal-dependent and rely on divalent ions (typically Zn²⁺) to stabilize the carbanion intermediate formed by the substrate within the active site (Prasad et al., 2013; Michels and Fothergill-Gilmore, 2017). Class-II aldolases typically form homodimers with a total molecular weight of ~78 kDa, but can

also form tetramers and octamers (Cooper et al., 1996; Pegan et al., 2009). Class-I aldolases are typically found in higher eukaryotic organisms (e.g., plants and animals). In contrast, Class-II aldolases are more common in morphologically less complex eukaryotic organisms (e.g., protozoans, fungi, yeasts, and algae), in prokaryotes and in some archaea (Tunio et al., 2010; Prasad et al., 2013; Falcicchio et al., 2014; Michels and Fothergill- Gilmore, 2017). In this study, we mentioned Class-II aldolase for gatY or KbaY from bacteria, and their subunits, gatZ and kbaZ with chaperone-like function for the proper and stable folding of the aldolases.

L. 99 Not "Subsequently" better to say "In order to achieve this"

► **As you recommended, we replaced it (L. 111).**

L.106 Replace "in over 500h" with "after 500h"

► **We corrected it (L117)**

L.108 Serial transfer culture? Presumably cultures. Explain of from what to what

► **We rephrased the text (L. 120).**

L.123 Cra is also known as FruR and it is its role as the fructose-specific repressor which is important here. The authors should state that Cra is FruR and the delta67 is removing the Cra/FruR binding site and derepressing the fruBAK operon.

► **As you advised, we additionally described it (L137, 138).**

L.142 onwards please refer to the figures where these results are shown. The text is not easy to follow. None of their growth curves give the original BL21(DE3) for comparison.

► **As advised, we rearranged figure 2 and included the descriptions of corresponding results cited with figures (L156~L158). In this text, we introduced mutations to WT with a stable growth profile (Fig. 1B) and tried to compare the growth rate and lag time between the created mutant strains for the three sugars media not with WT.**

L.155 a phenotype e.g., growth on fructose is not reversed by growth on another sugar (which is a different phenotype).

► **As advised, we deleted it (L173).**

L.158 loss of a specific regulator (AgaR) removes a sugar-operon specific repression and not general catabolite repression (generally due to cAMP, CAP and EIIA of the PTS)

► **As advised, we corrected it (L177)**

L.160 Their nomenclature of the strains carrying reconstructed mutations in BL21 is not orthodox but I appreciate their attempt to produce simple names for multimutated strains. The use of capital letters for the genes mutated and for the number of genes mutated is confusing and it might be easier if they use lowercase letters for the in the single, double, triple (e.g. triple mutation tCFA). In theory the s, d and t are redundant with the number of letters in the name (e.g. CFA is a triple mutation).

► **As you advised, we replaced upper case with lower case to indicate the number of mutated genes throughout the text (L160~L217).**

L.163 the growth curves for TCFA and TCFA(PTC) Fig. S2F are very different. Any comment?

Surprisingly some of the combinations of mutations do affect growth on glucose.

► Yes, as you mentioned, there are differences in growth curves of tCFA and tCFA^{PTC} stains in fructose media (S2F). We did not try further analysis to compare the effects of premature termination codon (PTC) and whole gene deletion in the repressor-coding gene. To our speculation based on these data, there are AT-rich regions in upstream and downstream of the *agaR* coding region. We believe that PTC mutation has more advantageous in genomic stability than deletion of *agaR*. It is currently unknown what effect the wild-type of AgaR (269 amino acids) causes in cells when reduced to 28 amino acids. In any way, there are large error bars in both two mutant strains (tCFA and tCFA^{PTC}) in D-Fru and D-Glc media, which clearly emphasizes both mutant types provoke perturbation in metabolic flux when sugar uptake.

L.181reduced to that of the original.... The sentence is badly constructed.

► We rephrased the relevant text (L198).

The mutation in the T7 RNAP promoter changed the -10 of the promoter from (... TATAAT) of the high expression *lacUV5* promoter to .. (TATGTT) corresponding to the original *lac* promoter and should thus have reduced expression of the T7 polymerase and of its downstream targets. (Was this verified?)

► Yes, the relevant part is also mentioned in the paper on *lacUV5* (Tabor et al. Proc Nat Acad Sci 1985; Noel et al. J Biol Chem 2000; Dubendorf et al. J Mol Biol 1991), which is consistent with our qRT-PCR data in Fig. 3A. The mRNA expression level of T7RNAP decreased in YF2 and YF1^{BL_GatY} strains with mutations in T7 RNAP promoter compared to dCF strain when these strains cultured in media supplemented with D-Tag (both 0.05% Fru+0.45% Tag and 0.5% Tag media) or other sugars. Also, the unpublished SDS-PAGE result (see below) shows that the expression level of BL_GatY in YF1 strains (2) seems to have decreased.

L. 186 Fig. S4A does not seem to show lag times. Please verify that the labels on the abscissa are correct.

► To clarify this, we re-depicted Fig. S4A.

What does "except for D-Fru in a plasmid-dependent manner" refer to? Do they mean "did not exhibit a lag time on glucose or fructose, except for Y1 in the absence of the *GatY* plasmid"??

► Two optimized strains (tCFA and tCFA^{PTC}) with no lag time on D-Tag and D-Glc (Fig. 2I and Fig. S4A) showed lags when cells were grown on D-Fru with and without the pET_BL_GatY plasmid (Fig. S2F and Fig. S4B). As you can see in (Fig. S4B), other recombinant strains harboring pET_BL_GatY plasmid showed lag on D-Fru due to a physiological burden by heterologous expression. To clarify this, we replaced the relevant sentence (L204).

L.198 a verb is missing - "was >2-fold higher"

► We revised the text (L215).

L.211 Fig.S5C Why should deletion of *fruK* (specific for fructose-1P) reduce growth on glucose in YF2?

► Generally, D-Glc and D-Fru catabolic pathways are highly linked. It is uncertain, but there has been reported that FruK can cause a conformation change of transcriptional regulator Cra through physical contact interaction (Singh et al. BioRxiv 2017). Therefore, it is considered that the absence of FruK may have delayed glucose availability by increasing the amount of the normal conformation of Cra, which inhibits the expression of glucose operon. This remains to further validate the effect of the bacterial fructose-1p kinase (FruK) on glucose availability at the molecular level.

L.213 "located in GalN PTS and galactitol PTS" Do they mean that *kbaY* is required for the metabolism of sugars transported by two different PTS. The *kbaY* gene is located with genes for the galactitol PTS (Fig.1A).

► We corrected wrong information (L230); As shown in Fig. 1A and Fig. S5B, the *kbaY* gene is located in the GalN PTS only, and not in the galactitol PTS.

L. 220 It is only novel in *E. coli* but a similar pathway exists already in *Salmonella* and *Klebsiella*!

► As we described above, *Salmonella* and *Klebsiella* have independent *tag* operons (*tagKTH*) along with the *fru* genes (STM2204, STM2205). We changed the term "novel" to "new route" (L129).

L.230 Fig. 3D What is the bar diagram to the right? Fig.3F What is the diagram to the right?

► As you pointed out, we added the missing information about the bar diagram in Fig. 3D and 3F (L806-L811).

L.260 fits well

► We revised it (L277).

L.271 we need some preliminary information. Please say that arabinose isomerase can catalyze the formation of tagatose from galactose (ref). They take advantage of the fact that BL21(DE3) is missing genes of the Leloir pathway to show cloned arabinose isomerase can allow growth on galactose via the tagatose utilization pathway.

► As you advised, we included additional information on L-arabinose isomerase (AI) that can catalyze the formation of D-Tag from D-Gal (Lee et al. AEM 2004) (L290-291).

L.282 what do they mean "grow solely by enzymatic activity"?

► This means that the platform host can grow on D-Gal only through AI activity for D-Gal to D-Tag. We also included an additional explanation to clarify this (L299).

L. 291 YFI (BL_GAT/EC_Ara ML) Their strain contain two pET plasmids with different antibiotic resistances but are they the same origin of replication? In which case the copy numbers of the two high copy plasmids are going to vary. 50 microg/ml of ampicillin will soon be used up in any case.

► Yes, we also concerned about the compatibility of two pET plasmids since the two plasmids are

of the same ori. In our case, we tested the pET_Duet system, but in the confirmation of expression through SDS-PAGE in a time-dependent manner, we concluded that the two plasmids system was much more effective for expression than the Duet plasmid system (see below). Furthermore, we used the two types of antibiotics, 50 µg/ml of ampicillin and 25 µg/ml of kanamycin, and culture transfers were conducted to fresh media with the presence of essential elements for growth; it was believed to prevent segregation of both plasmids.

Fig. 4D and 4F why is the lag for use of galactose so much longer in 4F than 4D? Fig. 4D should have the AraA plasmid indicated like the araA mutation in 4C.

► Generally, it is known that random mutagenesis yield less than 0.1% of positive selection among clones (Uchiyama and Miyazaki 2009; DeCastro et al. 2016). AraA has a homohexameric structure. Due to its structural characteristics, it is difficult to obtain positive mutants when using random mutagenesis compared to general enzymes. Even if the activity of AraA is improved, the reactivity of AraA to galactose is still marginal. In fact, the doubling time of selected mutants is close to 20 hours (revised Fig. 4G). Therefore, the growth-based selection needs a long period of time for colonization and/or dominance of a minor population of positive mutants.

L. 414 "without antibiotics, and an inducer" reads as if it is without antibiotics but with an inducer. I suppose they mean "without antibiotics and IPTG" i.e. without the inducer

► As you recommended, we revised it (L430).

L.555 "mutation(s) were introduced" it is unlikely they get a single mutation

► We corrected it (L570).

L.570 when cells HAD grown

► We corrected it (L585).

L.571 I do not understand why "growth was not faster than the growth rate of the original culture"?

► As you pointed out, we rephrased it (L.586).

L.734 what is the "clinker strategy"?

► clinker is a pipeline for generating a gene cluster map. We deleted the unnecessary word "strategy," and we have mentioned and cited the paper (Ref.45) in material methods (L. 760).

L.752 filled grey circle

► **We corrected it (L778).**

L.764 Analysis of the effects of

► **We corrected it (L787).**

L.783 Fig. 4B Why is the abscissa labelled as time on a histogram showing specific activity?

► **We intended to show that although L-AI can catalyze both sugars, L-Ara and D-Gal, as substrates, the activity of the enzyme to the substrate D-Gal is significantly slower than that of L-Ara.**

November 20, 2022

Prof. Dong-Woo Lee
Yonsei University
Biotechnology
Yonseiro 50
Seoul 03722
Korea (South), Republic of

Re: Spectrum03660-22R1 (A retro-aldol reaction prompted the evolvability of the phosphotransferase system to utilize rare sugars)

Dear Prof. Dong-Woo Lee:

Link Not Available

Sincerely,

Jing Han

Journals Department
Reviewer comments:

Reviewer #1 (Comments for the Author):

The author has studied reviewer's comments and suggestions carefully, and has made revision which marked in red in the revised paper. Regarding Reviewer #2 question L. 291, although it is unreasonable to use the two pET plasmids with the same ori, it will not affect the key results, so in summary, there is no question on the article now.

Reviewer #2 (Comments for the Author):

Rereading this paper I still find it difficult to follow and it is a shame that the authors do not have access to a competent English speaker who can understand and clearly present their results to a wide audience. I suppose it will be understandable to an interested specialist.

In general, although grammatically correct, the English sentence construction is not good and thus not easy to follow.

They have produced a detailed rebuttal to my numerous questions and suggestions to the first version but have made only minimal changes to the actual text, apparently they do not think other readers will have similar questions. E.g. they explain in the rebuttal the advantages of their system for constructing a tagatose-utilizing strain for selection of AI mutations compared to other published methods but do not explain that to the reader.

The authors have taken into account my linguistic corrections but I'm afraid that there are still a range of problems, in particular with the use of articles (when to use A or THE or neither). E.g. The presence of A (non-specific) rare sugar is different from the presence of THE (specific) rare sugar.

Compare "A system using A rare sugar" (e.g. tagatose, psicose, etc) with "A system using THE rare sugar, tagatose"

L. 30 of the fructose PTS (not all peripheral PTS are affected)

L. 77 environment enriched in A rare sugar (or THE rare sugar tagatose)

L. 79 using A rare sugar

L. 86 tagatose A naturally occurring rare sugar

L. 87 possess (present tense)

L. 89 not identified? Tag PTS genes are not found in E. coli; maybe they mean characterized?

L.110 the authors say that retro-aldol reaction is the reverse of the aldol condensation but have not equated it to the aldolase enzyme reaction.

I suggest in the title of the first result paragraph (L.83) should be

The retro-aldol reaction, carried out by an exogenous aldolase enzyme, initiated.....

L.124-7 cut this long sentence into three sentences. We chose cells as an endpoint strain. BL_GatY promoted... rich in the rare sugar, tagatose. Endogenous FruB and FruA which show >30% identity to enteric bacterial tagatose PTS counterparts,,;"

L. 129 it is not availability, rather ability "the ability to metabolize tagatose as promoted by the endogenous genes.....

L. 156 growth profile with little error? They have added this but I do not know to what it refers

L.175likely due to loss of AgaR repression of the kbaZagaVWA and agaSkbaYagaBCDI operons increasing expression of the aldolase kbaY and/or the aldolase chaperon, kbaZ .

NB 1 Verify the order and names of these genes (kbaZ, aldolase chaperone, is first gene of the aga operon according to EcoCyc and there is no agaF.

NB 2 Fig. 3A seems to show no up regulation of kbaZ! So they could just write "increasing expression of kbaY".

L. 230 kbaY showing very high sequence identity with aldolase of enteric bacteria? To what are they referring for the sequence identity? KbaY is an aldolase of BL21, an enteric bacteria. Are they referring to a difference in sequence with the B. licheniformis kbaY which is in a tagatose operon?

L. 290 as a candidate to be a D-galactose isomerase, which after mutagenesis might be responsible for the conversion of galactose to tagatose.

They might like to explain that any tagatose produced from galactose by AI needs to be phosphorylated at both 1 and 6 positions to produce the substrate for the KbaY aldolase. They presumably assume that the 1 position could be phosphorylated by FruA, if it is acting intracellularly (and not part of the PTS membrane transporter) thus allowing the second phosphorylation by the FruK A39S allele they have isolated.

L.273 Why do they cite Supp Fig. 6A and B, with NMR spectra of Fru1P, here? Is this relevant to the failed synthesis of Tag1P?? Did they synthesis Fru1P? but not Fru6P and Tag6P?

L.500 Did they really measure their 96 well plates with an Ultra8000 spectrophotometer, which seems to be a standard dual beam spectrometer and not a plate reader?

Fig. 1C They could say here that BL21(DE3) is also missing genes for galactose catabolism as well as the tagatose uptake

machinery.

Fig. 2E L.783 needs a capital E

Fig. 3A, 4E they should say what their RT-PCR results were normalized to.

Fig. 3D legend: L.802 the explanation they added concerning the bars next to the plate depictions indicating relative abundance of E. coli wt and mutant strains, needs a colour code.

Fig.4 L.810 Indirect D-tag catabolic pathway starting from D-gal

Fig.4D presumably they mean "supplemented with 0.5% ara OR 0.05% Fru and 0.45% Gal"

L.817 pET-22b (remove extra P)

Staff Comments:

Preparing Revision Guidelines

Please return the manuscript within 60 days; if you cannot complete the modification within this time period, please contact me. If you do not wish to modify the manuscript and prefer to submit it to another journal, please notify me of your decision immediately so that the manuscript may be formally withdrawn from consideration by Microbiology Spectrum.

The author has studied reviewer's comments and suggestions carefully, and has made revision which marked in red in the revised paper. Regarding Reviewer #2 question L. 291, although it is unreasonable to use the two pET plasmids with the same ori, it will not affect the key results, so in summary, there is no question on the article now.

Responses to reviewers (Spectrum03660-22R1)

□ Reviewer #1 (Comments for the Author):

The author has studied reviewer's comments and suggestions carefully, and has made revision which marked in red in the revised paper. Regarding Reviewer #2 question L. 291, although it is unreasonable to use the two pET plasmids with the same ori, it will not affect the key results, so in summary, there is no question on the article now.

► **We thank you so much for your kind appreciation of our work. Yes, we were also concerned about the use of two plasmids with the same ori. However, in response to the point raised by reviewer #2, we tested the pET_Duet system during the early stages of this study, and found that a two plasmid system was more effective for expression of the two genes (*gatY* and *araA*) than the Duet plasmid system. Accordingly, we used two types of antibiotics to retain the compatibility of both plasmids sharing the same ori.**

□ Reviewer #2 (Comments for the Author):

Rereading this paper I still find it difficult to follow and it is a shame that the authors do not have access to a competent English speaker who can understand and clearly present their results to a wide audience. I suppose it will be understandable to an interested specialist. In general, although grammatically correct, the English sentence construction is not good and thus not easy to follow.

► **As advised, the text has been revised throughout the manuscript by a professional editing service (www.bioedit.com).**

They have produced a detailed rebuttal to my numerous questions and suggestions to the first version but have made only minimal changes to the actual text, apparently they do not think other readers will have similar questions.

E.g. they explain in the rebuttal the advantages of their system for constructing a tagatose-utilizing strain for selection of AI mutations compared to other published methods but do not explain that to the reader.

► **As advised, we have revised the text to address the issues raised previously by the reviewers in the following areas.**

(L107–112) Regarding not including both subunits for the aldolase activity assay.

(L119–121) Regarding the selection of 0.05% fructose as a minimal helper substrate.

(L229–231) The mutation effect of the T7 RNAP promoter on T7RNAP.

(L331–334) The reason for the longer lag in Figure 4F than Figure 4D on galactose.

(L393–400) The reason for using an *E. coli* BL21 strain lacking *galTKE*, and an example of another strain with a weaker promoter than the strong T7lac promoter for expression.

(L421–425) The advantages of our system using a tagatose-utilizing strain for AI evolution.

The authors have taken into account my linguistic corrections but I'm afraid that there are still a range of problems, in particular with the use of articles (when to use A or THE or neither). E.g. The presence of A (non-specific) rare sugar is different from the presence of THE (specific) rare sugar.

Compare "A system using A rare sugar" (e.g. tagatose, psicose, etc) with " A system using THE rare sugar, tagatose"

► **The grammatical and syntax errors have been corrected by a professional English editing service throughout the revised manuscript.**

L. 30 of the fructose PTS (not all peripheral PTS are affected)

► **This has been revised (L30–31).**

L. 77 environment enriched in A rare sugar (or THE rare sugar tagatose)

L. 79 using A rare sugar

L. 86 tagatose A naturally occurring rare sugar

L. 87 possess (present tense)

► **As advised, we have corrected these (L77, L79, L87, L89, L132, L373).**

L. 89 not identified? Tag PTS genes are not found in *E. coli*; maybe they mean characterized?

► **We meant that D-Tag PTS genes from pathogenic enteric bacteria and *Bacillus licheniformis* were expressed in *E. coli* and functionally identified (Refs. #22 (Shakeri-Garakani *et al.* 2004) and #23 (Van der Heiden E *et al.* 2015)). To clarify this, we have rephrased the relevant text in the revised manuscript (L89–L91).**

L.110 the authors say that retro-aldol reaction is the reverse of the aldol condensation but have not equated it to the aldolase enzyme reaction.

► **As advised, we have corrected this (L116).**

I suggest in the title of the first result paragraph (L.83) should be *The retro-aldol reaction, carried out by an exogenous aldolase enzyme, initiated.....*

► **As advised, we have corrected this (L83).**

L.124-7 cut this long sentence into three sentences. We chose cells as an endpoint strain. *BL_GatY* promoted... rich in the rare sugar, tagatose. Endogenous *FruB* and *FruA* which show >30% identity to enteric bacterial tagatose PTS counterparts,

► **As advised, we have corrected this (L131).**

L. 129 it is not availability, rather ability "the ability to metabolize tagatose as promoted by the endogenous genes....."

► **As advised, we have corrected this (L135–136).**

L. 156 growth profile with little error? They have added this but I do not know to what it refers

► **We have deleted this (L160).**

L.175likely due to loss of *AgaR* repression of the *kbaZagaVWA* and *agaSkbaYagaBCDI* operons increasing expression of the aldolase *kbaY* and/or the aldolase chaperon, *kbaZ*.

NB 1 Verify the order and names of these genes (*kbaZ*, aldolase chaperone, is first gene of the *aga* operon according to *EcoCyc* and there is no *agaF*).

NB 2 Fig. 3A seems to show no up regulation of *kbaZ*! So they could just write "increasing expression of *kbaY*".

► **In fact, *E. coli* MG1655 (K-12 strain) does not possess *agaE* and *agaF* genes, according to *EcoCyc*. However, both genes are present in *E. coli* BL21(DE3) (NC_012971.2). As advised, we have corrected and revised the relevant text (L183–185).**

L. 230 *kbaY* showing very high sequence identity with aldolase of enteric bacteria? To what are they referring for the sequence identity? *KbaY* is an aldolase of BL21, an enteric bacteria. Are they referring to a difference in sequence with the *B. licheniformis* *kbaY* which is in a tagatose operon?

► We have included sequence identity values in the revised text (L243–245)

L. 290 as a candidate to be a D-galactose isomerase, which after mutagenesis might be responsible for the conversion of galactose to tagatose. They might like to explain that any tagatose produced from galactose by AI needs to be phosphorylated at both 1 and 6 positions to produce the substrate for the KbaY aldolase. They presumably assume that the 1 position could be phosphorylated by FruA, if it is acting intracellularly (and not part of the PTS membrane transporter) thus allowing the second phosphorylation by the FruK A39S allele they have isolated.

► Our comprehensive data indicated that intracellular D-gal is converted to D-tag by L-AI expressed in *E. coli* BL21(DE3), and, in turn, D-tag is phosphorylated to yield either D-tag-1P or D-tag-6P by fructose-specific PTS EIIAB components of membrane-anchored FruA (EIIBC) and cytosolic FruB directed towards the intracellular side.

It has been reported that PfkA and PfkB (also known as Fru-6P kinase) are involved in phosphorylation of Tag-6-P to generate Tag-1,6-BP (J. Babul *J. Biol. Chem.* 1978). However, our unpublished data below demonstrated that the essential components for D-tag catabolism are not PfkAB, but rather FruKBA (unpublished data C and D). Likewise, wild-type BL21(DE3) harboring AI and BL_GatY plasmids did not grow on galactose. Only the BL21(DE3) mutant strain with mutations in fructose-related genes and harboring the AI plasmid grew on D-gal, indicating that PfkA and PfkB are not relevant to supporting growth on D-tag. Accordingly, it is plausible that initiation of phosphorylation and D-tag catabolism are related to FruBKA not only via extracellular uptake of tagatose, but also through intracellularly produced tagatose by AI.

Unfortunately, this is not yet clear because we failed to obtain tag-1P to determine the primary substrate for FruK_A39S among tag-1P and tag-6P. Thus, we have re-depicted Figs. 1C, 3C, and 4A to reflect the fact that the specific position initially phosphorylated is unclear.

(Unpublished data) Comparison of growth profiles for the *E. coli* K-12 BW25113 strain harboring the pBAD-BL_GatY plasmid (A) vs. the *E. coli* BL21 (DE3) strain harboring the pET-BL_GatY plasmid (B) in M9 medium containing D-Fru as the sole carbon source. (C & D) The effect of single gene deletion in various transporters and PTS-related genes on the growth of *gatY*-expressing *E. coli* BW15113 and BL21(DE3) strains in M9 medium containing D-Tag as the sole carbon source.

L.273 Why do they cite Supp Fig. 6A and B, with NMR spectra of Fru1P, here? Is this relevant to the failed synthesis of Tag1P?? Did they synthesis Fru1P? but not Fru6P and Tag6P?

▶ **Yes, we synthesized D-Fru 1-P (L571) and obtained D-Fru 6-p and D-Tag 6-P from Sigma. As advised, we have moved Fig. S6 to the Materials and methods section (L542) and renamed 'Fig. S8' in the Supplementary information (L50–75).**

L.500 Did they really measure their 96 well plates with an Ultra8000 spectrophotometer, which seems to be a standard dual beam spectrometer and not a plate reader?

▶ **We have corrected this (L530).**

Fig. 1C They could say here that BL21(DE3) is also missing genes for galactose catabolism as well as the tagatose uptake machinery.

▶ **As advised, we have corrected this (L808).**

Fig. 2E L.783 needs a capital E

▶ **We have corrected this (L826).**

Fig. 3A, 4E they should say what their RT-PCR results were normalized to.

▶ **As advised, we have corrected this (L843 and L864).**

Fig. 3D legend: L.802 the explanation they added concerning the bars next to the plate depictions indicating relative abundance of E. coli wt and mutant strains, needs a colour code.

▶ **As advised, we have corrected this (L846–849).**

Fig.4 L.810 Indirect D-tag catabolic pathway starting from D-gal

Fig.4D presumably they mean "supplemented with 0.5% ara OR 0.05% Fru and 0.45% Gal"

▶ **We have corrected this (L857 and 862).**

L.817 pET-22b (remove extra P)

▶ **We have corrected this (L865).**

January 2, 2023

Prof. Dong-Woo Lee
Yonsei University
Biotechnology
Yonseiro 50
Seoul 03722
Korea (South), Republic of

Re: Spectrum03660-22R2 (A retro-aldol reaction prompted the evolvability of a phosphotransferase system for the utilization of rare sugars)

Dear Prof. Dong-Woo Lee:

Link Not Available

Sincerely,

Jing Han

Journals Department
Reviewer comments:

Reviewer #2 (Comments for the Author):

The authors have made more changes to their text in this second revision although it seems they have not always understood my queries.

I cannot say I'm impressed with the English editing service. I recommended that the paper be read by "a competent English speaker who can understand and clearly present their results to a wide audience". i.e. capable of correctly explaining their strategy and results. The Editing service basically checks for grammatical correctness and not whether the sentences are well constructed and that their strategy is logically explained and easily comprehensible.

I do not notice that much of their text has changed other than my corrections.

Each time I read it there are several more sentences, which to my mind, could be much better expressed. I've noted some more below with my suggested improvements. I could find many more.

There is a serious problem introduced with a previous correction:

Abstract L. 30-32. The authors have replaced "peripheral PTS" by "GlcNAc and Fru PTS" so that it now reads as if deletions in the Cra protein or its binding site are affecting the AgaR controlled genes ! They mean "deletions in the agaR gene and in the Cra binding site of the fruBKA operon" but do not make this clear and in the Abstract this is important.

What I think the authors want to say is

"These E. coli mutants lost tight regulation of both the N-acetyl-galactosamine and D-fructose PTS following a deletion in the agaR gene, encoding the GalNAc repressor, and a deletion in the binding site of the catabolite repressor/activator protein (Cra) upstream of the fruBKA operon, respectively".

This actually is frequent problem: they talk in generalities when their data is for a specific case. This is part of the problem of the use of A and THE that I mentioned previously.

In the title they write evolvability of a (i.e one) PTS for utilization of rare sugars (plural) but they have only demonstrated their system for one rare sugar, tagatose. So to my mind it should be "a rare sugar"

L.219 "introduction of kinase mutations" (is plural) but they have just one mutation in one kinase gene.

"Introduction of the FruK_A39S mutation to strains with mutations affecting regulation of the Fru PTS and AgaR controlled aldolase, further increased growth rates.

L. 55 "Most bacterial cells possess an individual phosphotransferase system (PTS) (9-11)" reads as if "each cell has a single/individual PTS" and not "Most bacterial cells have PTS for the individual sugars".

They could write

"Most bacteria possess one or several PTS systems for different sugars, producing phosphorylated sugars, which feed into glycolysis, in reactions which do not require oxygen, and which are followed by aerobic and/or....."

L.152 "downstream genes" relative to what? fruBKA are not downstream to cra but downstream of a Cra binding site.

"Differential growth profiles of the mutants indicated impaired Cra regulation of fruBKA, suggesting that the fruBKA gene products could be responsible for the ability of the mutant to use D-Tag for growth."

L180 "strains grew" (not were grown) (NB The strains were grown (by the experimenter, passive verb tense) on Tag but the strains grew (by themselves, active verb tense) on Tag with a lag of 150 h)

L245 induced (remove THE) sufficient expression of the fru operon

L251 "Novel metabolic flux was the result of a re-tuning of gene expression pathways provoked by the deletion of key regulatory elements."

L.252 The low sequence identity (of what? genes of the fruBKA operon, and authentic Tag transporter and D-Tag-1P or 6P kinase (from B. licheniformis or Klebsiella ?))

The low sequence identity does not reveal the loss of a regulated sugar specific PTS.

Maybe what they are trying to say is:

"Despite the low sequence identity of the genes of the fruBKA operon and authentic Tag transporter and D-Tag-1P or 6P kinase from B. licheniformis or Klebsiella, the loss of the tight regulation of the fru operon has allowed the Fru PTS to efficiently transport and phosphorylate D-Tag."

L.385 They mean deletion of Cra, the catabolite repressor/activator, binding sites. 'Catabolite repressor' sounds as if they mean CRP! This whole sentence is confusing.

"Loss of the Cra (catabolite repressor/activator) binding sites upstream of the fru operon, during adaptive evolution, is advantageous, compared to loss of the cra gene itself, because it allows adaption of the FruPTS to use a new ketohehexose, which does not act as an inducing signal."

L. 415 "Al-dependent growth on non-utilizable D-Gal, in the ALE-induced D-Tag auxotrophic strain"

L437 A cut-off (remove N)

Staff Comments:

Preparing Revision Guidelines

Please return the manuscript within 60 days; if you cannot complete the modification within this time period, please contact me. If you do not wish to modify the manuscript and prefer to submit it to another journal, please notify me of your decision immediately so that the manuscript may be formally withdrawn from consideration by Microbiology Spectrum.

Responses to reviewers (Spectrum03660-22R2)

□ Reviewer #2 (Comments for the Author):

The authors have made more changes to their texts in this second revision although it seems they have not always understood my queries. I cannot say I'm impressed with the English editing service. I recommended that the paper be read by "a competent English speaker who can understand and clearly present their results to a wide audience". i.e. capable of correctly explaining their strategy and results. The Editing service basically checks for grammatical correctness and not whether the sentences are well constructed and that their strategy is logically explained and easily comprehensible. I do not notice that much of their text has changed other than my corrections. Each time I read it there are several more sentences, which to my mind, could be much better expressed. I've noted some more below with my suggested improvements. I could find many more.

► **We greatly appreciate your kind help and critical comments. Based on your suggestions, we revised the texts marked in red. In addition to what you pointed out, we perused our manuscript and edited it to be easily comprehensible and clearly delivered to a wide audience.**

There is a serious problem introduced with a previous correction:

Abstract L. 30-32. The authors have replaced "peripheral PTS" by "GlcNAc and Fru PTS" so that it now reads as if deletions in the Cra protein or its binding site are affecting the AgaR controlled genes ! They mean "deletions in the agaR gene and in the Cra binding site of the fruBKA operon" but do not make this clear and in the Abstract this is important.

What I think the authors want to say is "These E. coli mutants lost tight regulation of both the N-acetyl-galactosamine and D-fructose PTS following a deletion in the agaR gene, encoding the GalNAc repressor, and a deletion in the binding site of the catabolite repressor/activator protein (Cra) upstream of the fruBKA operon, respectively".

► **As advised, we have corrected this (L31-34)**

This actually is frequent problem: they talk in generalities when their data is for a specific case. This is part of the problem of the use of A and THE that I mentioned previously. In the title they write evolvability of a (i.e one) PTS for utilization of rare sugars (plural) but they have only demonstrated their system for one rare sugar, tagatose. So to my mind it should be "a rare sugar"

► **We have corrected this (L1 and L16)**

L.219 "introduction of kinase mutations" (is plural) but they have just one mutation in one kinase gene. "Introduction of the FruK_A39S mutation to strains with mutations affecting regulation of the Fru PTS and AgaR controlled aldolase, further increased growth rates.

► **As advised, we have corrected this (L222)**

L. 55 "Most bacterial cells possess an individual phosphotransferase system (PTS) (9-11)" reads as if "each cell has a single/individual PTS" and not "Most bacterial cells have PTS for the individual sugars".

They could write "Most bacteria possess one or several PTS systems for different sugars, producing phosphorylated sugars, which feed into glycolysis, in reactions which do not require oxygen, and which are followed by aerobic and/or....."

► **As advised, we have corrected this (L61-62).**

L.152 "downstream genes" relative to what? fruBKA are not downstream to cra but downstream of a Cra binding site. "Differential growth profiles of the mutants indicated impaired Cra regulation of fruBKA, suggesting that the fruBKA gene products could be responsible for the ability of the mutant to use D-Tag for growth."

► **As advised, we have corrected this (L155-156).**

L180 "strains grew" (not were grown) (NB The strains were grown (by the experimenter, passive verb tense) on Tag but the strains grew (by themselves, active verb tense) on Tag with a lag of 150 h)

► **As advised, we have corrected this (L183)**

L245 induced (remove THE) sufficient expression of the fru operon

▶ **As advised, we have corrected this (L251).**

L251 "Novel metabolic flux was the result of a re-tuning of gene expression pathways provoked by the deletion of key regulatory elements."

▶ **As advised, we have corrected this (L254-255).**

L.252 The low sequence identity (of what? genes of the fruBKA operon, and authentic Tag transporter and D-Tag-1P or 6P kinase (from *B. licheniformis* or *Klebsiella* ?))

The low sequence identity does not reveal the loss of a regulated sugar specific PTS.

Maybe what they are trying to say is:

"Despite the low sequence identity of the genes of the fruBKA operon and authentic Tag transporter and D-Tag-1P or 6P kinase from *B. licheniformis* or *Klebsiella*, the loss of the tight regulation of the fru operon has allowed the Fru PTS to efficiently transport and phosphorylate D-Tag."

▶ **As advised, we have corrected this (L255-258)**

L.385 They mean deletion of Cra, the catabolite repressor/activator, binding sites. 'Catabolite repressor' sounds as if they mean CRP! This whole sentence is confusing.

"Loss of the Cra (catabolite repressor/activator) binding sites upstream of the fru operon, during adaptive evolution, is advantageous, compared to loss of the cra gene itself, because it allows adaptation of the FruPTS to use a new ketohexose, which does not act as an inducing signal."

▶ **As advised, we have corrected this (L390-392)**

L. 415 "AI-dependent growth on non-utilizable D-Gal, in the ALE-induced D-Tag auxotrophic strain"

▶ **As advised, we have corrected this (L419-420)**

L437 A cut-off (remove N)

▶ **As advised, we have corrected this (L442)**

January 25, 2023

Prof. Dong-Woo Lee
Yonsei University
Biotechnology
Yonseiro 50
Seoul 03722
Korea (South), Republic of

Re: Spectrum03660-22R3 (A retro-aldol reaction prompted the evolvability of a phosphotransferase system for the utilization of a rare sugar)

Dear Prof. Dong-Woo Lee:

Your manuscript has been accepted, and I am forwarding it to the ASM Journals Department for publication. You will be notified when your proofs are ready to be viewed.

Sincerely,

Jing Han
Editor, Microbiology Spectrum
